# Infinitely wide limits for deep Stable neural networks: sub-linear, linear and super-linear activation functions

**Alberto Bordino**                                     *alberto.bordino@warwick.ac.uk*
*University of Warwick*

**Stefano Favaro**                                     *stefano.favaro@edu.unito.it*
*University of Torino and Collegio Carlo Alberto*

**Sandra Fortini**                                     *sandra.fortini@unibocconi.it*
*Bocconi University*

**Reviewed on OpenReview:** *https://openreview.net/forum?id=A5tIluhDW6*

## Abstract

There is a growing literature on the study of large-width properties of deep Gaussian neural networks (NNs), i.e. deep NNs with Gaussian-distributed parameters or weights, and Gaussian stochastic processes. Motivated by some empirical and theoretical studies showing the potential of replacing Gaussian distributions with Stable distributions, namely distributions with heavy tails, in this paper we investigate large-width properties of deep Stable NNs, i.e. deep NNs with Stable-distributed parameters. For sub-linear activation functions, a recent work has characterized the infinitely wide limit of a suitable rescaled deep Stable NN in terms of a Stable stochastic process, both under the assumption of a "joint growth" and under the assumption of a "sequential growth" of the width over the NN's layers. Here, assuming a "sequential growth" of the width, we extend such a characterization to a general class of activation functions, which includes sub-linear, asymptotically linear and super-linear functions. As a novelty with respect to previous works, our results rely on the use of a generalized central limit theorem for heavy tails distributions, which allows for an interesting unified treatment of infinitely wide limits for deep Stable NNs. Our study shows that the scaling of Stable NNs and the stability of their infinitely wide limits may depend on the choice of the activation function, bringing out a critical difference with respect to the Gaussian setting.

## 1 Introduction

Deep (feed-forward) neural networks (NNs) play a critical role in many domains of practical interest, and nowadays they are the subject of numerous studies. Of special interest is the study of prior distributions over the NN's parameters or weights, namely random initializations of NNs. In such a context, there is a growing interest on large-width properties of deep NNs with Gaussian-distributed parameters, with emphasis on the interplay between infinitely wide limits of such NNs and Gaussian stochastic processes. Neal (1996) characterized the infinitely wide limit of a shallow Gaussian NN. In particular, let: i) $\boldsymbol{x} \in \mathbb{R}^d$ be the input of the NN; ii) $\tau : \mathbb{R} \to \mathbb{R}$ be an activation function; iii) $\theta = \{w_i^{(0)}, w, b_i^{(0)}, b\}_{i \geq 1}$ be the collection of NN's parameters such that $w_{i,j}^{(0)} \stackrel{d}{=} w_j \stackrel{iid}{\sim} N(0, \sigma_w^2)$ and $b_i^{(0)} \stackrel{d}{=} b \stackrel{iid}{\sim} N(0, \sigma_b^2)$ for $\sigma_w^2, \sigma_b^2 > 0$, with $N(\mu, \sigma^2)$ being the Gaussian distribution with mean $\mu$ and variance $\sigma^2$. Then, consider a rescaled shallow Gaussian NN defined as

$$f_{\boldsymbol{x}}(n)[\tau, n^{-1/2}] = b + \frac{1}{n^{1/2}} \sum_{j=1}^{n} w_j \tau(\langle w_j^{(0)}, \boldsymbol{x} \rangle_{\mathbb{R}^d} + b_j^{(0)}), \tag{1}$$

with $n^{-1/2}$ being the scaling factor. Neal (1996) showed that, as $n \to +\infty$ the NN output $f_{\boldsymbol{x}}(n)[\tau, n^{-1/2}]$ converges in distribution to a Gaussian random variable (RV) with mean zero and a suitable variance. The proof follows by an application of the Central Limit Theorem (CLT), thus relying on minimal assumptions on $\tau$, as it is sufficient to ensure that $\mathbb{E}[(g_j(\boldsymbol{x}))^2]$ is finite, where $g_j(\boldsymbol{x}) = w_j \tau(\langle w_j^{(0)}, \boldsymbol{x} \rangle_{\mathbb{R}^d} + b_j^{(0)})$. The result of Neal (1996) has been extended to a general input matrix, i.e. $k > 1$ inputs of dimension $d$, and deep Gaussian NNs, assuming both a "sequential growth" (Der & Lee, 2005) and a "joint growth" (de G. Matthews et al., 2018) of the width over the NN's layers. See Theorem A.1. In general, all these large-width asymptotic results rely on some minimal assumptions for the function $\tau$, thus allowing to cover the most popular activation functions.

Neal (1996) first discussed the problem of replacing the Gaussian distribution of the NN's parameters with a Stable distribution, namely a distribution with heavy tails (Samorodnitsky & Taqqu, 1994), leaving to future research the study of infinitely wide limits of Stable NNs. In a recent work, Favaro et al. (2020; 2022a) characterized the infinitely wide limit of deep Stable NNs in terms of a Stable stochastic process, assuming both a "joint growth" and a "sequential growth" of the width over the NN's layers. Critical to achieve the infinitely wide Stable process is the assumption of a sub-linear activation function $\tau$, i.e. $|\tau(x)| \le a + b|x|^\beta$, with $a, b > 0$ and $0 < \beta < 1$. In particular, for a shallow Stable NN, let: $\boldsymbol{x} \in \mathbb{R}^d$ be the input of the NN; ii) $\tau : \mathbb{R} \to \mathbb{R}$ be the sub-linear activation function of the NN; iii) $\theta = \{w_i^{(0)}, w, b_i^{(0)}, b\}_{i \ge 1}$ be the NN's parameters such that $w_{i,j}^{(0)} \stackrel{d}{=} w_j \stackrel{iid}{\sim} S_\alpha(\sigma_w)$ and $b_i^{(0)} \stackrel{d}{=} b \stackrel{iid}{\sim} S_\alpha(\sigma_b)$ for $\alpha \in (0, 2]$ and $\sigma_w, \sigma_b > 0$, with $S_\alpha(\sigma)$ being the symmetric Stable distribution with stability $\alpha$ and scale $\sigma$. Then, consider the rescaled shallow Stable NN

$$f_{\boldsymbol{x}}(n)[\tau, n^{-1/\alpha}] = b + \frac{1}{n^{1/\alpha}} \sum_{j=1}^{n} w_j \tau(\langle w_j^{(0)}, \boldsymbol{x} \rangle_{\mathbb{R}^d} + b_j^{(0)}), \tag{2}$$

with $n^{-1/\alpha}$ being the scaling factor. The NN (1) is recovered from (2) by setting $\alpha = 2$. Favaro et al. (2020) showed that, as $n \to +\infty$ the NN output $f_{\boldsymbol{x}}(n)[\tau, n^{-1/\alpha}]$ converges in distribution to a Stable RV with stability $\alpha$ and a suitable scale. See Theorem A.2 in the Appendix. Differently from the Gaussian setting of (de G. Matthews et al., 2018), the result of Favaro et al. (2020) relies on the assumption of a sub-linear activation function. This is a strong assumption, as it does not allow to cover some popular activation functions.

The use of Stable distribution for the NN's parameters, in place of Gaussian distributions, was first motivated through empirical analyses in Neal (1996), which show that while all Gaussian weights vanish in the infinitely wide limit, some Stable weights retain a non-negligible contribution, allowing to represent "hidden features". See also Der & Lee (2005) and Lee et al. (2022), and references therein, for an up-to-date discussion on random initializations of NNs with classes of distributions beyond the Gaussian distribution. In particular, in the context of heavy tails distributions, Fortuin et al. (2022) showed that wide Stable (convolutional) NNs trained with gradient descent lead to a higher classification accuracy than Gaussian NNs. Still in such a context, Favaro et al. (2022b) considered a Stable NN with a ReLU activation function, showing that the large-width training dynamics of the NN is characterized in terms of kernel regression with a Stable random kernel, in contrast with the well-known (deterministic) neural tangent kernel in the Gaussian setting Jacot et al. (2018); Arora et al. (2019). In general, the different behaviours between the Gaussian setting and the Stable settings arise from the large-width sample path properties of the NNs, as shown in Favaro et al. (2022a;b), which make $\alpha$-Stable NNs more flexible than Gaussian NNs. See Figure 1 and Figure 2 in the Appendix.

## 1.1 Our contributions

In this paper, we investigate the large-width asymptotic behaviour of deep Stable NNs with a general activation function. Given $f : \mathbb{R} \to \mathbb{R}$ and $g : \mathbb{R} \to \mathbb{R}$ we write $f(z) \simeq g(z)$ for $z \to +\infty$ if $\lim_{z \to +\infty} f(z)/g(z) = 1$, and $f(z) = \mathcal{O}(g(z))$ for $z \to +\infty$ if if there exists $C > 0$ and $z_0$ such that $f(z)/g(z) \le C$ for every $z \ge z_0$. Analogously for $z \to -\infty$. We omit the explicit reference to the limit of $z$ when there is no ambiguity or when the relations hold both for $z \to +\infty$ and for $z \to -\infty$. Now, let $\tau : \mathbb{R} \to \mathbb{R}$ be a continuous functions

and define:

$$E_1 = \{\tau \in \mathcal{C}(\mathbb{R}; \mathbb{R}) : |\tau(z)| = \mathcal{O}(|z|^\beta) \text{ with } 0 \le \beta < 1\};$$

$$E_2 = \{\tau \in \mathcal{C}(\mathbb{R}; \mathbb{R}) : |\tau(z)| \simeq |z|^\gamma \text{ and } \tau \text{ strictly increasing for } |z| > a, \text{ for some } \gamma, a > 0\};$$

$$E_3 = \{\tau \in \mathcal{C}(\mathbb{R}; \mathbb{R}) : \tau(z) \simeq z^\gamma \text{ for } z \to +\infty, |\tau(z)| = \mathcal{O}(|z|^\beta) \text{ with } \beta < \gamma \text{ for } z \to -\infty$$
$$\text{and } \tau \text{ strictly increasing for } z > a, \text{ for some } \gamma, a > 0\}.$$

We characterize the infinitely wide limits of shallow Stable NNs with activation functions in $E_1$, $E_2$ and $E_3$, assuming a $d$-dimensional input. Such a characterization is then applied recursively to derive the behaviour of a deep Stable NNs under the simplified setting of "sequential growth", i.e. when the hidden layers grow wide one at the time. Our results extends the work of Favaro et al. (2020; 2022a) to a general asymptotically linear function, i.e. $E_2 \cup E_3$ choosing $\gamma = 1$, and super-linear functions, i.e. $E_2 \cup E_3$ choosing $\gamma > 1$. As a novelty with respect to previous works, our results rely on the use of a generalized CLT (Uchaikin & Zolotarev, 2011; Otiniano & Gonçalves, 2010), which reduces the characterization of the infinitely wide limit of a deep Stable NNs to the study of the tail behaviour of a suitable transformation of Stable random variables. This allows for a unified treatment of infinitely wide limits for deep Stable NNs, providing an alternative proof of the result of Favaro et al. (2020; 2022a) under the class $E_1$. Our results show that the scaling of a Stable NN and the stability of its infinitely wide limits depend on the choice of the activation function, thus bringing out a critical difference with respect to the Gaussian setting. While in the Gaussian setting the choice of $\tau$ does not affect the scaling $n^{-1/2}$ required to achieve the Gaussian process, in the Stable setting the use of an asymptotically linear function results in a change of the scaling $n^{-1/\alpha}$, through an additional $(\log n)^{-1/\alpha}$ term, to achieve the Stable process. Such a phenomenon was first observed in Favaro et al. (2022b) for a shallow Stable NN with a ReLU activation function, which is indeed an asymptotically linear activation function.

## 1.2 Organization of the paper

Section 2 contains the main results of the paper: i) the weak convergence of a shallow Stable NN with an activation function $\tau$ in the classes $E_1$, $E_2$ and $E_3$, for an input $x = 1$ and no biases; ii) the weak convergence of a deep Stable NN with an activation function $\tau$ in the classes $E_1$, $E_2$ and $E_3$, for an input $x \in \mathbb{R}^d$ and biases. In Section 3 we discuss some natural extensions of our work, as well as some directions for future research.

## 2 Main results

Let $(\Omega, \mathcal{H}, \mathbb{P})$ be a generic probability space on which all the RVs are assumed to be defined. Given a RV $Z$, we define its cumulative distribution function (CDF) as $P_Z(z) = \mathbb{P}(Z \le z)$, its survival function as $\overline{P}_Z(z) = 1 - P_Z(z)$, and its the density function with respect to the Lebesgue measure as $p_Z(z) = \frac{dP_Z(z)}{dz}$, using the notation $P_Z(dz)$ to indicate $p_Z(z)dz$. A RV $Z$ is symmetric if $Z \overset{d}{=} -Z$, i.e. if $Z$ and $-Z$ have the same distribution, that is $\overline{P}_Z(z) = P_Z(-z)$ for all $z \in \mathbb{R}$. We say that $Z_n$ converges to $Z$ in distribution, as $n \to +\infty$, if for every point of continuity $z \in \mathbb{R}$ of $P_Z$ it holds $P_{Z_n}(z) \to P_Z(z)$ as $n \to +\infty$, in which case we write $Z_n \overset{d}{\longrightarrow} Z$. Given $f : \mathbb{R} \to \mathbb{R}$ and $g : \mathbb{R} \to \mathbb{R}$ we write $f(z) = o(g(z))$ for $z \to +\infty$ if $\lim_{z \to +\infty} f(z)/g(z) = 0$. Analogously for $z \to -\infty$. As before, we omit the reference to the limit of $z$ when there is no ambiguity or when the relation holds for both $z \to +\infty$ and for $z \to -\infty$. Recall that a measurable function $L : (0, +\infty) \to (0, +\infty)$ is called slowly varying at $+\infty$ if $\lim_{x \to +\infty} L(ax)/L(x) = 1$ for all $a > 0$.

**Definition 2.1.** *A $\mathbb{R}$-valued RV $X$ has Stable distribution with stability $\alpha \in (0, 2]$, skewness $\beta \in [-1, 1]$, scale $\sigma > 0$ and shift $\mu \in \mathbb{R}$, and we write $X \sim S_\alpha(\sigma, \beta, \mu)$, if its characteristic function is $\varphi_X(t) = \mathbb{E}[e^{itX}] = e^{\psi(t)}$, for $t \in \mathbb{R}$, where*

$$\psi(t) = \begin{cases} -\sigma^\alpha |t|^\alpha [1 + i\beta \tan(\frac{\alpha\pi}{2}) sign(t)] + i\mu t & \alpha \ne 1 \\ -\sigma |t| [1 + i\beta \frac{2}{\pi} sign(t) \log(|t|)] + i\mu t & \alpha = 1. \end{cases}$$

By means of Samorodnitsky & Taqqu (1994, Property 1.2.16), if $X \sim S_\alpha(\sigma, \beta, \mu)$ with $0 < \alpha < 2$ then $\mathbb{E}[|X|^r] < +\infty$ for $0 < r < \alpha$, and $\mathbb{E}[|X|^r] = +\infty$ for any $r \geq \alpha$. A $\mathbb{R}$-valued RV $X$ is distributed as the symmetric $\alpha$-Stable distribution with scale parameter $\sigma$, and we write $X \sim S_\alpha(\sigma)$, if $X \sim S_\alpha(\sigma, 0, 0)$, which implies that $\varphi_X(t) = \mathbb{E}[e^{itX}] = e^{-\sigma^\alpha |t|^\alpha}$, $t \in \mathbb{R}$. This allows to prove that if $X \sim S_\alpha(\sigma)$, then $aX \sim S_\alpha(|a|\sigma)$; see Samorodnitsky & Taqqu (1994, Property 1.2.3). Furthermore, one has the following complete characterization of the tail behaviour of the CDF and PDF of Stable RVs: for a symmetric $\alpha$-Stable distribution, Samorodnitsky & Taqqu (1994, Proposition 1.2.15) states that, if $X \sim S_\alpha(\sigma)$ with $0 < \alpha < 2$,

$$\overline{P}_X(x) = P_X(-x) \simeq \frac{1}{2} C_\alpha \sigma^\alpha |x|^{-\alpha},$$

where

$$C_\alpha = \left( \int_0^{+\infty} x^{-\alpha} \sin(x) dx \right)^{-1} = \frac{2}{\pi} \Gamma(\alpha) \sin\left( \alpha \frac{\pi}{2} \right) = \begin{cases} \frac{1-\alpha}{\Gamma(2-\alpha)\cos(\pi\frac{\alpha}{2})} & \alpha \neq 1 \\ \frac{2}{\pi} & \alpha = 1. \end{cases}$$

As before, if $X \sim S_\alpha(\sigma)$ with $0 < \alpha < 2$, then $p_X(x) = p_X(-x) \simeq (\alpha/2)C_\alpha \sigma^\alpha |x|^{-\alpha-1}$ for $x \to +\infty$ holds true.

For an activation function belonging to the classes $E_1$, $E_2$ and $E_3$, we characterize the infinitely wide limit of a deep Stable NN, assuming a $d$-dimensional input and a "sequential growth" of the width over the NN's layers. Critical is the use of the following generalized CLT (Uchaikin & Zolotarev, 2011; Otiniano & Gonçalves, 2010).

**Theorem 2.1** (Generalized CLT). *Let $Z$ be a RV such that $\overline{P}_Z(z) \simeq cz^{-p}L(z)$ and $P_Z(-z) \simeq dz^{-p}L(z)$ for some $c, d > 0$, $0 < p < 2$ and with $L$ being a slow varying function. Moreover, let $(Z_n)_{n \geq 1}$ be a sequence of RVs iid as $Z$. If*

$$a_n = \begin{cases} 0 & 0 < p < 1 \\ (c - d) \log n & p = 1 \\ \mathbb{E}[Z] & 1 < p < 2, \end{cases}$$

*then, as $n \to +\infty$*

$$\frac{1}{(nL(n))^{1/p}} \sum_{i=1}^n (Z_i - a_n) \xrightarrow{d} S_p\left( \left[ \frac{c+d}{C_p} \right]^{\frac{1}{p}}, \frac{c-d}{c+d}, 0 \right). \tag{3}$$

For $p < 1$, no centering turns out to be necessary in (3), due to the "large" normalizing constants $n^{-1/p}$, which smooth out the differences between the right and the left tail of $Z$. For $p > 1$, the centering in (3) is the common one, namely the expectation. The case $p = 1$ is a special case: the expectation does not exist, so it cannot be used as a centering in (3); on the other hand, centering is necessary for convergence because the normalizing constant $n^{-1}$ does not grow sufficiently fast to smooth the differences between the right and the left tail of $Z$. In particular, the term including $\log n$ comes from the asymptotic behaviour of truncated moments.

## 2.1 Shallow Stable NNs: large-width asymptotics for an input $x = 1$ and no biases

We start by considering a shallow Stable NN, for an input $x = 1$ and no biases. Let $w^{(0)} = [w_1^{(0)}, w_2^{(0)}, \dots]^T$ and $w = [w_1, w_2, \dots]^T$ independent sequences of RVs such that $w_j^{(0)} \overset{iid}{\sim} S_{\alpha_0}(\sigma_0)$ and $w_j \overset{iid}{\sim} S_{\alpha_1}(\sigma_1)$. Then, we set $Z_j = w_j \tau(w_j^{(0)})$, where $\tau : \mathbb{R} \to \mathbb{R}$ is a continuous non-decreasing function, and define the shallow Stable NN

$$f(n)[\tau, p] = \frac{1}{n^{1/p}} \sum_{j=1}^n Z_j \tag{4}$$

with $p > 0$. From the definition of the shallow Stable NN (4), being $Z_1, Z_2, \dots$ iid according to a certain RV $Z$, it is sufficient to study the tail behaviour of $\overline{P}_Z(z)$ and $P_Z(-z)$ in order to obtain the convergence

in distribution of $f(n)[\tau, p]$. As a general strategy, we proceed as follows: i) we study the tail behaviour of $X \cdot \tau(Y)$ where $X \sim S_{\alpha_x}(\sigma_x)$, $Y \sim S_{\alpha_x}(\sigma_y)$, $X \perp\!\!\!\perp Y$ and $\tau \in E_1$, $\tau \in E_2$ and $\tau \in E_3$; ii) we make use of the generalized CLT, i.e. Theorem 2.1, in order to characterize the infinitely wide limit of the shallow Stable NN (4).

Note that to find the tail behaviour of $X \cdot \tau(Y)$ it is sufficient to find the tail behaviour of $|X \cdot \tau(Y)|$, and then use the fact that $\mathbb{P}[X \cdot \tau(Y) > z] = (1/2) \cdot \mathbb{P}[|X \cdot \tau(Y)| > z]$ for every $z \geq 0$, since $X \cdot \tau(Y)$ is symmetric as $X$ is so. Then, to find the asymptotic behaviour of the survival function of $|X \cdot \tau(Y)|$ we make use of some results in the theory of convolution tails and domain of attraction of Stable distributions. Hereafter, we recall some basic facts. Given two CDFs $F$ and $G$, the convolution $F * G$ is defined as $F * G(t) = \int F(t - y)dG(y)$, which inherits the linearity of the convolution and the commutativity of the convolution from properties of the integral operator. Recall that a function $F$ on $[0, +\infty]$ has exponential tails with rate $\alpha$ ($F \in \mathcal{L}_\alpha$) if and only if

$$\lim_{t \to +\infty} \frac{\overline{F}(t - y)}{\overline{F}(t)} = e^{\alpha y}, \quad \text{for all real } y \in \mathbb{R}.$$

Then,

$$\bar{F}(t) = a(t) \exp\left[-\int_0^t \alpha(v)dv\right], \quad \text{where } a(t) \to a > 0, \alpha(t) \to \alpha, \text{ as } t \to +\infty.$$

A complimentary definition is the following: a function $U$ on $[0, +\infty]$ is regularly varying with exponent $\rho$ ($U \in \mathcal{RV}_\rho$) if and only if

$$\lim_{t \to +\infty} \frac{U(yt)}{U(t)} = y^\rho, \quad \text{for all } y > 0.$$

Then,

$$U(t) = a(t) \exp\left[\int_0^t \frac{\rho(v)}{v}dv\right], \quad \text{where } a(t) \to a > 0, \rho(t) \to \rho, \text{ as } t, \to +\infty,$$

i.e. the Karamata's representation of $U$. Clearly $F \in \mathcal{L}_\alpha$ if and only if $\bar{F}(\ln t) \in \mathcal{RV}_{-\alpha}$. The next lemma provides the tail behaviour of the convolution of $F$ and $G$, assuming that they have exponential tails with the same rates.

**Lemma 2.2** (Theorem 4 of Cline (1986)). *Let $F, G \in \mathcal{L}_\alpha$ for some $\alpha > 0$, $f \in \mathcal{RV}_\beta$ and $g \in \mathcal{RV}_\gamma$ where $f(t) = e^{\alpha t}\overline{F}(t)$ and $g(t) = e^{\alpha t}\overline{G}(t)$ and $\beta > -1$ and $\gamma > -1$. Then*

$$\overline{F * G}(t) \simeq \frac{\Gamma(1 + \beta)\Gamma(1 + \gamma)}{\Gamma(1 + \beta + \gamma)} \alpha t e^{\alpha t}\overline{F}(t)\overline{G}(t), \quad \text{as } t \to +\infty.$$

We make use of Lemma 2.2 to find the tail behaviour of $|X \cdot \tau(Y)|$ when $|X|$ and $|\tau(Y)|$ have regularly varying truncated CDFs with same rates. If $|X|$ and $|\tau(Y)|$ have regularly varying truncated CDFs with different rates, then we make use of the next lemma, which describes the tail behaviour of $U \cdot W$, where $U$ and $W$ are two independent non-negative RVs such that $\mathbb{P}[U > u]$ is regularly varying of index $-\alpha \leq 0$ and $\mathbb{E}[W^\alpha] < +\infty$.

**Lemma 2.3.** *Suppose $U$ and $W$ are two independent non-negative RVs such that $\mathbb{E}[W^\alpha] < +\infty$ and $\mathbb{P}[W > u] = o(\mathbb{P}[U > u])$. If $\mathbb{P}[U > u] \simeq cu^{-\alpha}$, with $c > 0$, then $\mathbb{P}[UW > u] \simeq \mathbb{E}[W^\alpha] \cdot \mathbb{P}[U > u]$.*

Lemma 2.3 was stated in Breiman (1965) for $\alpha \in [0, 1]$, and then extended by Cline & Samorodnitsky (1994) for all values of $\alpha$, still under the hypothesis that $\mathbb{E}[W^{\alpha+\epsilon}] < +\infty$ for some $\epsilon > 0$. Lemma 2.3 provides a further extension in the case $\mathbb{P}[U > x] \simeq cx^{-\alpha}$, with $c > 0$, and has been proved in Denisov & Zwart (2005). Based on Lemma 2.2 and Lemma 2.3, it remains to find the tail behaviour of $|X|$ and $|\tau(Y)|$. For the former, it is easy to show that $\overline{P}_{|X|}(t) := \mathbb{P}[|X| > t] \simeq C_{\alpha_x}\sigma_x^{\alpha_x}t^{-\alpha_x}$, while, for the latter, we have the next lemma.

**Lemma 2.4** (Tail behaviour of $\tau(Y)$, $\tau \in E_2 \cup E_3$). *Assuming $Y \sim S_\alpha(\sigma)$, then: i) $\mathbb{P}[|\tau(Y)| > t] \simeq C_\alpha\sigma^\alpha t^{-\alpha/\gamma}$ if $\tau \in E_2$; ii) $\mathbb{P}[|\tau(Y)| > t] \simeq \frac{1}{2}C_\alpha\sigma^\alpha t^{-\alpha/\gamma}$ if $\tau \in E_3$.*

*Proof.* If $\tau(z)$ is strictly increasing for $z > a$ and $\tau(z) \simeq z^\gamma$ for $z \to +\infty$ with $\gamma > 0$, then $\tau^{-1}(y) \simeq y^{1/\gamma}$. Analogously at $-\infty$. We refer to Theorem 5.1 of Olver (1974) for the case $\gamma = 1$. Now, starting with $\tau \in E_2$ and defining the inverse of $\tau$ where the activation is strictly increasing, we can write for a sufficiently large $t$:

$$P(|\tau(Y)| > t) = P(\tau(Y) > t) + P(\tau(Y) < -t) \simeq \frac{1}{2}C_\alpha\sigma^\alpha|\tau^{-1}(t)|^{-\alpha} + \frac{1}{2}C_\alpha\sigma^\alpha|\tau^{-1}(-t)|^{-\alpha}$$

$$\simeq \frac{1}{2}C_\alpha\sigma^\alpha t^{-\alpha/\gamma} + \frac{1}{2}C_\alpha\sigma^\alpha|t|^{-\alpha/\gamma} = C_\alpha\sigma^\alpha t^{-\alpha/\gamma}.$$

Instead, if $\tau \in E_3$, then there exits $b > 0$ and $y_0 < 0$ such that $|\tau(y)| < b|y|^\beta$ for $y \leq y_0$. Then, for $t$ sufficiently large,

$$P(|\tau(Y)| > t, Y < 0) \leq P(b|Y|^\beta > t, Y < 0) \simeq \frac{1}{2}C_\alpha\sigma^\alpha(bt)^{-\alpha/\beta}.$$

Furthermore,

$$P(|\tau(Y)| > t, Y > 0) = P(Y > \tau^{-1}(t)) \simeq \frac{1}{2}C_\alpha\sigma^\alpha|\tau^{-1}(t)|^{-\alpha} \simeq \frac{1}{2}C_\alpha\sigma^\alpha t^{-\alpha/\gamma},$$

hence, since $\beta < \gamma$, it holds that $P(|\tau(Y)| > t) \simeq \frac{1}{2}C_\alpha\sigma^\alpha t^{-\alpha/\gamma}$, which concludes the proof. $\quad\square$

Based on the previous results, it is easy to derive the tail behaviour of $|X \cdot \tau(Y)|$, which is stated in the next theorem.

**Theorem 2.5** (Tail behaviour of $|X \cdot \tau(Y)|$). *Let $|Z| = |X \cdot \tau(Y)|$ where $X$ and $Y$ are independent and distributed respectively as $S_{\alpha_x}(\sigma_x)$ and $S_{\alpha_y}(\sigma_y)$. If $\tau \in E_1$ and $\beta\alpha_x < \alpha_y$, then*

$$\overline{P}_{|Z|}(t) \simeq C_{\alpha_x}\sigma_x^{\alpha_x}\mathbb{E}[|\tau(Y)|^{\alpha_x}]t^{-\alpha_x}.$$

*For $\tau \in E_2 \cup E_3$, define $\underline{\alpha} = \min(\alpha_x, \alpha_y/\gamma)$ and $c_\tau = \frac{1}{2}$ if $\tau \in E_3$ and $c_\tau = 1$ otherwise. Then*

$$\overline{P}_{|Z|}(z) \simeq \begin{cases} c_\tau C_{\underline{\alpha}\gamma}\sigma_y^{\underline{\alpha}\gamma}\mathbb{E}[|X|^{\underline{\alpha}}]z^{-\underline{\alpha}} & \text{if} \quad \gamma > \alpha_y/\alpha_x \\ c_\tau C_{\underline{\alpha}}C_{\underline{\alpha}\gamma}\sigma_x^{\underline{\alpha}}\sigma_y^{\underline{\alpha}\gamma}z^{-\underline{\alpha}}\log z & \text{if} \quad \gamma = \alpha_y/\alpha_x \\ C_{\underline{\alpha}}\sigma_x^{\underline{\alpha}}\mathbb{E}[|\tau(Y)|^{\underline{\alpha}}]z^{-\underline{\alpha}} & \text{if} \quad \gamma < \alpha_y/\alpha_x. \end{cases}$$

*Proof.* We start from the proof of the first case, i.e. $\tau \in E_1$. Here, $|\tau(Y)| < b|Y|^\beta$ for certain $\beta \in (0, 1)$ and $b > 0$, when $|Y|$ is larger than some $y_0 > 0$, hence there exists $c > 0$ such that $\mathbb{E}[|\tau(Y)|^{\alpha_x}] \leq c + \mathbb{E}[b|Y|^{\beta\cdot\alpha_x}] < +\infty$, being $\beta\alpha_x < \alpha_y$ by hypothesis. The thesis then follows from Lemma 2.3. An analogous strategy can be used in the case $\alpha_x \neq \alpha_y/\gamma$. Indeed, $\mathbb{E}[|X|^{\alpha_y/\gamma}] < +\infty$ if $\alpha_x > \alpha_y/\gamma$ and $\mathbb{E}[|\tau(Y)|^{\alpha_x}] < +\infty$ if $\alpha_x < \alpha_y/\gamma$. Hence Lemma 2.3 allows to conclude. A different situation arises when $\alpha_x = \alpha_y/\gamma$. In this case, consider the RVs $\log|X|$ and $\log|\tau(Y)|$ and observe that, for $t > 0$,

$$\overline{P}_{\log|X|}(t) := \mathbb{P}[\log|X| > t] = \mathbb{P}[|X| > e^t] \simeq C_{\alpha_x}\sigma_x^{\alpha_x}e^{-\alpha_x t} \in \mathcal{L}_{\alpha_x},$$

i.e. $\mathbb{P}[\log|X| > t]$ has an exponential tail with index $\alpha_x$, and the same has $\overline{P}_{\log|\tau(Y)|} := \mathbb{P}[\log|\tau(Y)| > t]$ since $\alpha_x = \alpha_y/\gamma$. Furthermore, $e^{\alpha_x t} \cdot \overline{P}_{\log|X|}(t) \in \mathcal{RV}_0$ and $e^{\alpha_x t} \cdot \overline{P}_{\log|\tau(Y)|} \in \mathcal{RV}_0$, hence we apply Lemma 2.2 with $\beta = \gamma = 0$, and obtain that

$$\mathbb{P}[\log|X \cdot \tau(Y)| > t] = \mathbb{P}[\log|X| + \log|\tau(Y)| > t] = \overline{P}_{\log|X|} * \overline{P}_{\log|\tau(Y)|}(t)$$

$$= \alpha_x t e^{\alpha_x t}\overline{P}_{\log|X|}(t)\overline{P}_{\log|\tau(Y)|}(t)$$

$$= \alpha_x C_{\alpha_x}C_{\alpha_y}\sigma_x^{\alpha_x}\sigma_y^{\alpha_y}t e^{-\alpha_x t}.$$

It is sufficient to evaluate this expression in $\log t$ to obtain the thesis. As for the case $\tau \in E_3$, the proof is the same except for an extra $\frac{1}{2}$ in the tail behaviour of $\overline{P}_{|\tau(Y)|}(t)$. $\quad\square$

Based on Theorem 2.5, the next theorem is an application of the generalized CLT, i.e. Theorem 2.1, that provides the infinitely wide limit of the shallow Stable NN (4), with the activation function $\tau$ belonging to the classes $E_1, E_2, E_3$.

**Theorem 2.6** (Shallow Stable NN, $\tau \in E_1, E_2, E_3$)**.** *Consider $f(n)[\tau, p]$ defined in (4). If $\tau \in E_1$ and $\beta \alpha_1 < \alpha_0$, then*

$$f(n)[\tau, \alpha_1] \xrightarrow{d} S_{\alpha_1} \left( \sigma_1 \left( \mathbb{E}_{Z \sim S_{\alpha_0}(\sigma_0)}[|\tau(Z)|^{\alpha_1}] \right)^{1/\alpha_1} \right).$$

*If $\tau \in E_2 \cup E_3$, define $\underline{\alpha} = \min(\alpha_1, \alpha_0/\gamma)$, $c_\tau = \frac{1}{2}$ if $\tau \in E_3$ and $c_\tau = 1$ otherwise, and $m_n(\gamma) = \log n$ if $\gamma = \alpha_0/\alpha_1$ and $m_n(\gamma) = 1$ otherwise. Then*

$$m_n(\gamma)^{-1/\underline{\alpha}} f(n)[\tau, \underline{\alpha}] \xrightarrow{d} S_{\underline{\alpha}}(\sigma),$$

*where*

$$\sigma = \begin{cases} \sigma_0^\gamma \sigma_1 \left( c_\tau \frac{C_{\underline{\alpha}\gamma}}{C_{\underline{\alpha}}} \mathbb{E}_{Z \sim S_{\alpha_1}(1)}[|Z|^{\underline{\alpha}}] \right)^{1/\underline{\alpha}} & if \quad \gamma > \alpha_0/\alpha_1 \\ \sigma_0^\gamma \sigma_1 \left( c_\tau \underline{\alpha} C_{\gamma \underline{\alpha}} \right)^{1/\underline{\alpha}} & if \quad \gamma = \alpha_0/\alpha_1 \\ \sigma_1 \left( \mathbb{E}_{Z \sim S_{\underline{\alpha}}(\sigma_0)}[|\tau(Z)|^{\underline{\alpha}}] \right)^{1/\underline{\alpha}} & if \quad \gamma < \alpha_0/\alpha_1. \end{cases}$$

*Proof.* Observe that, since $w_j \cdot \tau(w_j^{(0)})$ is symmetric, then

$$\mathbb{P}[w_j \cdot \tau(w_j^{(0)}) > t] = \mathbb{P}[w_j \cdot \tau(w_j^{(0)}) < -t] = \frac{1}{2}\mathbb{P}[|w_j \cdot \tau(w_j^{(0)})| > t].$$

Hence, the proof of this theorem follows by combining Theorem 2.5 and the generalized CLT, i.e. Theorem 2.1, after observing that $a_n = 0$ due to the symmetry of $w_j \cdot \tau(w_j^{(0)})$. $\qquad \square$

The term $(\log n)^{-1/\underline{\alpha}}$ in the scaling in the case $\tau \in E_2 \cup E_3$ and $\alpha_1 = \alpha_0/\gamma$, is a novelty with respect to the Gaussian setting. That is, NNs with Gaussian-distributed parameters are not affected by the presence of one activation in place of another as the scaling is always $n^{-1/2}$, while this is not true for Stable NNs as shown above.

## 2.2 Deep Stable NNs: large-width asymptotics for an input $x \in \mathbb{R}^d$ and biases

The above results can be extended to deep Stable NNs, assuming a "sequential growth" of the width over the NN's layers, for an input $\boldsymbol{x} = (x_1, ..., x_d) \in \mathbb{R}^d$ and biases. Differently from the "joint growth", under which the widths of the layers growth simultaneously, the "sequential growth" implies that the widths of the layers growth one at a time. Because of the assumption of a "sequential growth", the study of the large width behaviour of a deep Stable NN reduces to a recursive application of Theorem 2.6. In particular, let $\theta = \{w_i^{(0)}, ..., w_i^{(L-1)}, w, b_i^{(0)}, ..., b_i^{(L-1)}, b\}_{i \geq 1}$ the set of all parameters and $\boldsymbol{x} \in \mathbb{R}^d$ be the input. Define $\forall i \geq 1$ and $\forall l = 1, ..., L-1$

$$\begin{cases} w_i^{(0)} = [w_{i,1}^{(0)}, w_{i,2}^{(0)}, \dots w_{i,d}^{(0)}] & \in \mathbb{R}^d \\ w_i^{(l)} := [w_{i,1}^{(l)}, w_{i,2}^{(l)}, \dots w_{i,n}^{(l)}] & \in \mathbb{R}^n \\ w = [w_1, w_2, \dots w_n] & \in \mathbb{R}^n \\ w_{i,j}^{(0)}, w_{i,j}^{(l)}, w_i, b_i^{(l)}, b & \in \mathbb{R} \\ w_{i,j}^{(0)} \stackrel{d}{=} w_{i,j}^{(l)} \stackrel{d}{=} w_j \stackrel{d}{=} b_i^{(l)} \stackrel{d}{=} b \stackrel{iid}{\sim} S_\alpha(1). \end{cases} \quad (5)$$

Then, we define the deep Stable NN as

$$
\begin{cases}
g_j^{(1)}(\boldsymbol{x}) = \sigma_w \langle w_j^{(0)}, \boldsymbol{x} \rangle_{\mathbb{R}^d} + \sigma_b b_j^{(0)} \\
g_j^{(l)}(\boldsymbol{x}) = \sigma_b b_j^{(l-1)} + \sigma_w \nu(n)^{-\frac{1}{\alpha}} \sum_{i=1}^n w_{j,i}^{(l-1)} \tau(g_i^{(l-1)}(\boldsymbol{x})), \quad \forall l = 2, ..., L \\
f_{\boldsymbol{x}}(n)[\tau, \alpha] = g_1^{(L+1)}(\boldsymbol{x}) = \sigma_b b + \sigma_w \nu(n)^{-\frac{1}{\alpha}} \sum_{j=1}^n w_j \tau(g_j^{(L)}(\boldsymbol{x})),
\end{cases}
\tag{6}
$$

where $\nu(n) = n \cdot \log(n)$ if $\tau \in E_2 \cup E_3$ with $\gamma = 1$ and $\nu(n) = n$ otherwise, and $\langle \cdot, \cdot \rangle_{\mathbb{R}^d}$ denotes the Euclidean inner product in $\mathbb{R}^d$. Note that the definition (6) coincides with the definition (4) provided that $L = 1$, $\sigma_b = 0$, $d = 1$ and $x = 1$. For the sake of simplicity and readability of the results, we have restricted ourselves to the case where all the parameters are Stable-distributed with same index $\alpha$, but this setting can be further generalized.

The next theorem provides the infinitely wide limit of the deep Stable NN (6), assuming a "sequential growth" of the width over the NN's layers. In particular, if we expand the width of the hidden layers to infinity one at the time, from $l = 1$ to $l = L$, then it is sufficient to apply Theorem 2.6 recursively through the NN's layers.

**Theorem 2.7** (Deep Stable NN, $\tau \in E_1$ and $\tau \in E_2 \cup E_3$ with $\gamma \leq 1$). *Consider $g_j^{(l)}(\boldsymbol{x})$ for fixed $j = 1, ..., n$ and $l = 2, ...., L+1$ as defined in (6). Then, as the width goes to infinity sequentially over the NN's layers, it holds*

$$
g_j^{(l)}(\boldsymbol{x}) \xrightarrow{d} S_\alpha(\sigma_x^{(l)}),
$$

*where $\sigma_x^{(1)} = (\sigma_w^\alpha \sum_{j=1}^d |x_j|^\alpha + \sigma_b^\alpha)^{1/\alpha}$, and, for $l = 2, \ldots, L+1$*

$$
\sigma_x^{(l)} = \begin{cases}
\left( \sigma_w^\alpha \mathbb{E}_{Z \sim S_\alpha(\sigma_x^{(l-1)})}[|\tau(Z)|^\alpha] + \sigma_b^\alpha \right)^{1/\alpha} & \text{if} \quad \tau \in E_1 \text{ or } \tau \in E_2 \cup E_3, \text{ with } \gamma < 1, \\
\left( c_\tau \alpha C_\alpha \sigma_w^\alpha (\sigma_x^{(l-1)})^\alpha + \sigma_b^\alpha \right)^{1/\alpha} & \text{if} \quad \tau \in E_2 \cup E_3, \text{ with } \gamma = 1,
\end{cases}
$$

*with $c_\tau = 1/2$ if $\tau \in E_3$ and $c_\tau = 1$ otherwise.*

*Proof.* The case $L = 1$ deals again with a shallow Stable NN but considering non-null Stable biases and a more complex type of input. The result follows from Theorem 2.6 by replacing $w_j^{(0)}$ with $g_j^{(1)}(\boldsymbol{x})$ and $\sigma_0$ with $\sigma_x = (\sigma_b^\alpha + \sigma_w^\alpha \sum_{j=1}^d |x_j|^\alpha)^{\frac{1}{\alpha}}$ thanks to the fact that $g_j(\boldsymbol{x}) \overset{iid}{\sim} S_\alpha(\sigma_x)$ for $j = 1, ..., n$. This can be easily proved using the following properties of the Stable distribution (Samorodnitsky & Taqqu, 1994, Chapter 1): i) if $X_1 \perp\!\!\!\perp X_2$ and $X_i \sim S_\alpha(\sigma_i)$ then $X_1 + X_2 \sim S_\alpha([\sigma_1^\alpha + \sigma_2^\alpha]^{\frac{1}{\alpha}})$; ii) if $c \neq 0$ and $X_1 \sim S_\alpha(\sigma_1)$ then $c \cdot X_1 \sim S_\alpha(|c|\sigma_1)$. The proof for the case $L > 1$ is based on the fact that the $g_i^{(l-1)}(\boldsymbol{x})$'s are independent and identically distributed as $S_\alpha(\sigma_x^{(l-1)})$ since they inherit these properties from the iid initialization of weights and biases: the thesis then follows applying the result for $L = 1$ layer after layer and substituting $\sigma_x^{(l-1)}$ in place of $\sigma_x$. $\square$

Theorem 2.7 includes the limiting behaviour of $f_{\boldsymbol{x}}(n)[\tau, \alpha]$ in the case $l = L+1$. It is possible to write an explicit form of the scale parameter by recursively expanding the scale parameters of the hidden layers. See Subsection 2.3 for an example in the case of the ReLU activation function. Before concluding, we point out that when using a sub-linear activation, i.e. $\tau \in E_1$ or $\tau \in E_2 \cup E_3$ with $\gamma \in (0,1)$, or a asymptotically linear activation, i.e. $\tau \in E_2 \cup E_3$ with $\gamma = 1$, the index $\alpha$ of the limiting Stable distribution does not change as the depth of a node increases so that, even for a very deep NN, the limiting output is distributed as a $\alpha$-Stable distribution. Such a behaviour is not preserved for super-linear activation functions, i.e. $\tau \in E_2 \cup E_3$ with $\gamma > 1$. When $\alpha_1 < \alpha_0/\gamma$, the convergence result of Theorem 2.6 involves a Stable RV with index equal to $\alpha_0/\gamma$, and not $\alpha_0$. In case $\alpha_x = \alpha_y = \alpha$, this is the case when $\gamma > 1$, which corresponds to a super-linear activation in $E_2 \cup E_3$. The fact that the limiting RV takes a factor $1/\gamma$ prevent us from writing a theorem in the setting of Definition (6) because we would not be able to apply the property i) above as it describes the distribution of the sum of independent Stable RVs with different scales but same index. We are then forced to adjust the initialization of the biases and to this purpose we define a new setting. Let

$\theta = \{w_i^{(0)}, ..., w_i^{(L-1)}, w, b_i^{(0)}, ..., b_i^{(L-1)}, b\}_{i \geq 1}$ the set of all parameters and $\boldsymbol{x} \in \mathbb{R}^d$ be the input. Define $\forall i \geq 1$ and $\forall l = 0, ..., L-1$

$$
\begin{cases}
w_i^{(0)} = [w_{i,1}^{(0)}, w_{i,2}^{(0)}, \ldots w_{i,d}^{(0)}] & \in \mathbb{R}^d \\
w_i^{(l)} = [w_{i,1}^{(l)}, w_{i,2}^{(l)}, \ldots w_{i,d}^{(l)}] & \in \mathbb{R}^n \\
w = [w_1, w_2, \ldots w_n] & \in \mathbb{R}^n \\
w_{i,j}^{(0)}, w_{i,j}^{(l)}, w_i, b_i^{(l)}, b & \in \mathbb{R} \\
w_{i,j}^{(l)} \stackrel{d}{=} w_j \stackrel{iid}{\sim} S_\alpha(1) \\
b_i^{(l)} \stackrel{iid}{\sim} S_{\frac{\alpha}{\gamma^l}}(1) \\
b \stackrel{iid}{\sim} S_{\frac{\alpha}{\gamma^L}}(1).
\end{cases}
\tag{7}
$$

Then, we define the deep Stable NN as

$$
\begin{cases}
g_j^{(1)}(\boldsymbol{x}) = \sigma_w \langle w_j^{(0)}, \boldsymbol{x} \rangle_{\mathbb{R}^d} + \sigma_b b_j^{(0)} \\
g_j^{(l)}(\boldsymbol{x}) = \sigma_b b_j^{(l-1)} + \sigma_w n^{-\frac{1}{\alpha}} \sum_{i=1}^n w_{j,i}^{(l-1)} \tau(g_i^{(l-1)}(\boldsymbol{x})), \quad \forall l = 2, ..., L \\
f_{\boldsymbol{x}}(n)[\tau, \alpha] = g_1^{(L+1)}(\boldsymbol{x}) = \sigma_b b + \sigma_w n^{-\frac{1}{\alpha}} \sum_{j=1}^n w_j \tau(g_j^{(L)}(\boldsymbol{x})).
\end{cases}
\tag{8}
$$

The next theorem provides the counterpart of Theorem 2.7 for the deep Stable NN (6). It provides the infinitely wide limit of the deep Stable NN (8), assuming a "sequential growth" of the width over the NN's layers.

**Theorem 2.8** (Deep Stable NN, $\tau \in E_2 \cup E_3$ with $\gamma > 1$). *Consider $g_j^{(l)}(\boldsymbol{x})$ for fixed $j = 1, ..., n$ and $l = 2, ...., L+1$ as defined in (8). As the width goes to infinity sequentially over the NN's layers,*

$$
g_j^{(l)}(\boldsymbol{x}) \xrightarrow{d} S_{\alpha/\gamma^{l-1}} \left( \sigma_x^{(l)} \right),
$$

*where $\sigma_x^{(1)} = (\sigma_w^\alpha \sum_{j=1}^d |x_j|^\alpha + \sigma_b^\alpha)^{1/\alpha}$, and*

$$
\sigma_x^{(l)} = \left( \frac{c_\tau C_{\alpha/\gamma^{l-2}}}{C_{\alpha/\gamma^{l-1}}} \sigma_w^{\alpha/\gamma^{l-1}} (\sigma_x^{(l-1)})^{\alpha/\gamma^{l-2}} \mathbb{E}_{Z \sim S_\alpha(1)}[|Z|^{\alpha/\gamma^{l-1}}] + \sigma_b^{\alpha/\gamma^{l-1}} \right)^{\gamma^{l-1}/\alpha},
$$

*with $c_\tau = 1/2$ if $\tau \in E_3$ and $c_\tau = 1$ otherwise.*

*Proof.* The proof is along lines similar to the proof of Theorem 2.7. Notice that the fact that $b_i^{(l-1)} \stackrel{iid}{\sim} S_{\alpha/\gamma^{l-1}}(1)$ is critical to conclude the proof. $\square$

As a corollary of Theorem 2.8, the limiting distribution of $f_{\boldsymbol{x}}(n)[\tau, \alpha]$, as $n \to +\infty$, follows a $(\alpha/\gamma^L)$-Stable distribution with scale parameter that can be computed recursively. That is, for a large number $L$ of layers, the stability parameter of the limiting distribution is close to zero. As we have pointed out for a shallow Stable NN, this is a peculiar feature of the class $\tau \in E_2 \cup E_3$ with $\gamma > 1$, and it can be object of a further analysis.

### 2.3 Some examples

As Theorem 2.6 is quite abstract, we present some concrete examples using well-known activation functions. First consider the case when $\tau = \tanh \in E_1$, since it is bounded. Then, the output of a shallow Stable NN (4) is such that

$$
f(n)[\tanh, \alpha_1] \xrightarrow{d} S_{\alpha_1} \left( \sigma_1 \left( \mathbb{E}_{Z \sim S_{\alpha_0}(\sigma_0)}[|\tanh(Z)|^{\alpha_1}] \right)^{1/\alpha_1} \right).
$$

See also Favaro et al. (2020). As for the new classes of activations introduced here, we can start considering the super-linear activation $\tau(z) = z^3 \in E_2$ with $\gamma = 3$, in the case of a shallow NN with $\alpha_1 = \alpha_0 = \alpha$ and obtain that

$$f(n)[(\cdot)^3, \alpha/3] \xrightarrow{d} S_{\alpha/3}\left(\sigma_0^3 \sigma_1 \left(\frac{C_\alpha}{C_{\alpha/3}} \mathbb{E}_{Z \sim S_\alpha(1)}[|Z|^{\alpha/3}]\right)^{3/\alpha}\right),$$

with the novelty here lying in the fact that the index of the limiting output is $\alpha/3$ instead of $\alpha$. As for asymptotically linear activations, if you take $\tau = id$, i.e. the identity function, again under the hypothesis of a shallow NN with $\alpha_1 = \alpha_0 = \alpha$, you obtain that

$$(\log n)^{-1/\alpha} f(n)[id, \alpha] \xrightarrow{d} S_\alpha\left(\left[\alpha C_\alpha\right]^{1/\alpha} \sigma_0 \sigma_1\right),$$

which shows the presence of an extra logarithmic factor of $(\log n)^{-1/\alpha_1}$ in the scaling for the first time. Beware that this behaviour, which is a critical difference with the Gaussian case, does not show up only with asymptotically linear activations: for example, if you take $\alpha_1 = 1$, i.e. Cauchy distribution, $\alpha_0 = 3/2$, i.e. Holtsmark distribution, and $\tau(z) = z^{3/2}$ then

$$(\log n)^{-1} f(n)[(\cdot)^{3/2}, 1] \xrightarrow{d} S_1\left(C_1 \sigma_0 \sigma_1\right).$$

Finally, we consider the ReLU activation function, which is one of the most popular activation functions. The following theorems deal with shallow Stable NNs and deep Stable NNs, respectively, with a ReLU activation function.

**Theorem 2.9** (Shallow Stable NN, ReLU). *Consider $Z = X \cdot ReLU(Y)$ where $X \sim S_\alpha(\sigma_x)$, $Y \sim S_\alpha(\sigma_y)$, $X \perp\!\!\!\perp Y$. Then*

$$\overline{P}_Z(z) = P_Z(-z) \simeq \frac{\alpha}{4} C_\alpha^2 \sigma_y^\alpha \sigma_x^\alpha z^{-\alpha} \log z.$$

*Furthermore, if $f(n)[ReLU, p] = n^{-\frac{1}{p}} \sum_{j=1}^n w_j ReLU(w_j^{(0)})$, where $ReLU(t) = \max\{0, t\}$ and $w_j^{(0)} \overset{iid}{\sim} S_\alpha(\sigma_0) \perp\!\!\!\perp w_j \overset{iid}{\sim} S_\alpha(\sigma_1)$, then*

$$(\log n)^{-\frac{1}{\alpha}} f(n)[\tau, \alpha] \xrightarrow{d} S_\alpha\left(\left[\frac{1}{2}\alpha C_\alpha\right]^{\frac{1}{\alpha}} \sigma_0 \sigma_1\right).$$

*Proof.* The theorem follows easily from Theorem 2.5 and Theorem 2.6, after noticing that ReLU $\in E_3$ with $\beta = 0$ and $\gamma = 1$. In addition to that, we also provide an alternative proof which can be useful in other applications. First, the distribution of $Q := \text{ReLU}(Y)$ is $\mathbb{P}[Q \leq q] = \mathbb{P}(\max\{0, Y\} \leq q) = \mathbb{P}[Y \leq q]\mathbb{1}\{q \geq 0\}$, from which we observe that $Q$ is neither discrete nor absolutely continuous with respect to the Lebesgue measure as it has a point mass of $\frac{1}{2}$ at $z = 0$ while the remaining $\frac{1}{2}$ of the mass is concentrated on $\mathbb{R}_+$ accordingly to the Stable law of $Y$ on $(0, +\infty)$. Hence, having in mind the shape of $P_Q(q) = \mathbb{P}[Q \leq q]$, we derive the approximation for the tails of the distribution of $X \cdot \text{ReLU}(Y)$ and, as usual, we make use of the generalized CLT to prove the following theorem. We prove the tail behaviour of $\overline{P}_Z(z)$ first. For any $z > 0$ we can write that

$$\overline{P}_Z(z) = \int_0^{+\infty} \mathbb{P}\left[Q > \frac{z}{x}\right] p_X(x) dx = \int_0^{+\infty} \mathbb{P}\left[Y > \frac{z}{x}\right] p_X(x) dx,$$

since $Y$ and $Q$ have the same distribution on $(0, +\infty)$. Now, observe that

$$\mathbb{P}[XY \geq z] = \int_{\mathbb{R}} \mathbb{P}[xY \geq z] P_X(dx) = 2 \int_0^{+\infty} \mathbb{P}\left[Y \geq \frac{z}{x}\right] p_X(x) dx$$

where the second equality holds by splitting the integral on $\mathbb{R}$ into the sum of the integrals on $(-\infty, 0)$ and $(0, +\infty)$ and using the fact that $Y$ is symmetric. It follows that, for every $z > 0$, $\overline{P}_Z(z) = \frac{1}{2}\mathbb{P}[XY \geq z]$. Applying the results for $\tau = id$, we find that

$$\overline{P}_Z(z) = \frac{1}{2}\mathbb{P}[XY \geq z] \simeq \frac{\alpha}{4} C_\alpha^2 \sigma_y^\alpha \sigma_x^\alpha z^{-\alpha} \log z.$$

The proof for the asymptotic behaviour of $P_Z(z)$ works in the same way after fixing $z < 0$ and using a change of variable while the convergence in distribution of $(\log n)^{-\frac{1}{\alpha}} f(n)[\tau, \alpha]$ follows by a direct application of the generalized CLT. $\qquad\square$

Theorem 2.9 can be extended to deep Stable NN with input $\boldsymbol{x} = (x_1, ..., x_d) \in \mathbb{R}^d$ and considering the biases.

**Theorem 2.10** (Deep Stable NN, ReLU). *Consider the deep Stable NN with ReLU activation defined as follows*

$$
\begin{cases}
g_j^{(1)}(\boldsymbol{x}) = \sigma_w \langle w_j^{(0)}, \boldsymbol{x} \rangle_{\mathbb{R}^d} + \sigma_b b_j^{(0)} \\
g_j^{(l)}(\boldsymbol{x}) = \sigma_b b_j^{(l-1)} + \sigma_w (n \log n)^{-\frac{1}{\alpha}} \sum_{i=1}^n w_{j,i}^{(l-1)} ReLU(g_i^{(l-1)}(\boldsymbol{x})), \forall l = 2, ..., L \\
f_{\boldsymbol{x}}(n)[ReLU, \alpha] = g_1^{(L+1)}(\boldsymbol{x}) = \sigma_b b + \sigma_w (n \log n)^{-\frac{1}{\alpha}} \sum_{j=1}^n w_j ReLU(g_j^{(L)}(\boldsymbol{x})).
\end{cases} \tag{9}
$$

*Then, under the hypothesis of Stable initialization for weights and biases as in (6), as the width of the previous layers goes to infinity sequentially,*

$$
g_j^{(l)}(\boldsymbol{x}) \xrightarrow{d} S_\alpha \left( \sigma_x^{(l)} \right),
$$

*where $\sigma_x^{(1)} = (\sigma_w^\alpha \sum_{j=1}^d |x_j|^\alpha + \sigma_b^\alpha)^{1/\alpha}$, and, for $l = 2, \ldots, L+1$,*

$$
\sigma_x^{(l)} = \left( \frac{1}{2} \alpha C_\alpha (\sigma_x^{(l-1)})^\alpha \sigma_w^\alpha + \sigma_b^\alpha \right)^{1/\alpha}.
$$

*Proof.* The proof is along lines similar to the proof of Theorem 2.7 with $\gamma = 1$ and $\tau \in E_3$. $\qquad\square$

Then, as a corollary of Theorem 2.10, the limiting distribution of $f_{\boldsymbol{x}}(n)[\text{ReLU}, \alpha]$, as $n \to +\infty$, is the distribution of a $\alpha$-Stable RV whose scale can be computed recursively. In particular, we can write the following statement.

**Corollary 2.10.1.** *Under the setting of Theorem 2.10 with a generic depth $L$,*

$$
f(n)[ReLU, \alpha] \xrightarrow{d} S_\alpha \left( \left[ \left( \frac{1}{2} \alpha C_\alpha \sigma_w^\alpha \right)^L \sigma_x^\alpha + \sum_{i=0}^{L-1} (\frac{1}{2} \alpha C_\alpha \sigma_w^\alpha)^i \sigma_b^\alpha \right]^{\frac{1}{\alpha}} \right).
$$

*Proof.* The claim is true for $L = 1$, which can be proved using the standard two properties of the Stable distribution. Moreover, for a NN with depth of $L+1$, using Theorem 2.10, the scale is

$$
\left[ \frac{1}{2} \alpha C_\alpha \sigma_w^\alpha \left[ \left( \frac{1}{2} \alpha C_\alpha \sigma_w^\alpha \right)^L \sigma_x^\alpha + \sum_{i=0}^{L-1} \left( \frac{1}{2} \alpha C_\alpha \sigma_w^\alpha \right)^i \sigma_b^\alpha \right] + \sigma_b^\alpha \right]^{\frac{1}{\alpha}}
$$

$$
= \left[ \left( \frac{1}{2} \alpha C_\alpha \sigma_w^\alpha \right)^{L+1} \sigma_x^\alpha + \sum_{i=0}^L \left( \frac{1}{2} \alpha C_\alpha \sigma_w^\alpha \right)^i \sigma_b^\alpha \right]^{\frac{1}{\alpha}},
$$

which concludes the proof. $\qquad\square$

## 3 Discussion

In a recent work, Favaro et al. (2020; 2022a) has characterized the infinitely wide limit of deep Stable NNs under the assumption of a sub-linear activation function. Here, we made use of a generalized CLT to characterize the infinitely wide limit of deep Stable NNs with a general activation function belonging to the classes $E_1$, $E_2$ and $E_3$. For $\alpha_1 = \alpha_0/\gamma$, and in particular for the choices $\tau = id$ and $\tau = \text{ReLU}$ with $\alpha_0 = \alpha_1$, Theorem 2.6 shows that the right scaling of the NN is $(n \log n)^{-1/\alpha}$, thus including the extra factor $(\log n)^{-1/\alpha}$ with respect to sub-linear activation functions. For $\alpha_1 > \alpha_0/\gamma$, and in particular for the choice

of a super-linear activations with $\alpha_0 = \alpha_1$, Theorem 2.8 shows that the distribution of the limiting output is $(\alpha_0/\gamma)$-Stable, with $\gamma > 1$, and this may have undesirable consequences for posterior estimates in case of a very deep NN. In general, our work brings out the critical role of the generalized CLT, which is not as popular as the classical CLT, in the study of the large width behaviour of deep Stable NNs. As the classical CLT plays a critical role in the study of the large-width behaviour of deep Gaussian NNs (Neal, 1996), our work shows how the generalized CLT plays the same critical role in the study of the large-width behaviour of deep Stable NNs.

A natural direction for future research consists in extending our results to deep Stable NNs with $k > 1$ inputs of dimension $d$, i.e. a $d \times k$ input matrix $\mathbf{X}$. A unified treatment of such a problem, would require a multidimensional versions of the generalized CLT, i.e. a CLT dealing with $k > 1$ dimensional Stable distributions, which is not available in the probabilistic/statistical literature. For a shallow Stable NN with a ReLU activation function, this problem has been considered in Favaro et al. (2022b), where the infinitely wide limit of the NN is characterized through a careful analysis of the large-width behaviour of the characteristic function of the NN. A further natural problem consists in extending our results to the case of a "joint growth" of the width over the NN's layers, i.e. the widths of the layers growth simultaneously (Favaro et al., 2022a). In general, under the setting specified in definition (6) or definition (8), one may consider a deep NN defined as follows:

$$f_i^{(1)}(\mathbf{X}, n) = \sum_{j=1}^{d} w_{i,j}^{(1)} \mathbf{x}_j + b_i^{(1)} \mathbf{1}^T$$

and

$$f_i^{(l)}(\mathbf{X}, n) = \frac{1}{f(n)^{1/\alpha}} \sum_{j=1}^{n} w_{i,j}^{(l)} \left( \tau \circ f_j^{(l-1)}(\mathbf{X}, n) \right) + b_i^{(l)} \mathbf{1}^T,$$

with $f_i^{(1)}(\mathbf{X}, n) = f_i^{(1)}(\mathbf{X})$, where $\mathbf{1}$ denotes the $k$-dimensional unit (column) vector, $\circ$ is the element-wise application and $f(n) = n \cdot \log n$ if $\tau \in E_2$ and $f(n) = n$ otherwise. Then, the goal consists in extending our results to $f_i^{(l)}(\mathbf{X}, n)$, assuming a "joint growth" of the width over the NN's layers. The case $\tau \in E_1$ was already tackled by Favaro et al. (2020; 2022a) but the other two cases are missing. In particular, Favaro et al. (2022a) showed that the assumptions of a "joint growth" and of a "sequential growth" lead to the same infinitely wide limit for a deep Stable NN with a sub-linear activation function. Instead, a critical difference between the assumption of a "joint growth" and the assumption of a "sequential growth" arises in the study of rate of convergence of the NN to its infinitely wide limit. In particular, Favaro et al. (2022a) investigated rates of convergence, in the sup-norm distance, for deep Stable NNs with a sub-linear activation function, showing that the assumption of a "joint growth" leads to a rate that depends on the depth, whereas the assumption of a "sequential growth" leads to a rate that is independent of the depth. We conjecture that an analogous phenomenon holds true for deep Stable NNs with linear and super-linear activation functions. In particular, we expect that the infinitely wide limits presented in our work hold true under the assumption that the width grows jointly over the layers, suggesting that a difference between the "joint growth" and the "sequential growth" may require the study of convergence rates. To study the large width asymptotic behaviour under the assumption of a "joint growth", it might be useful to use Theorem 1 of Fortini et al. (1997), which gives sufficient conditions for the convergence to a mixture of infinitely divisible laws: to apply this theorem, one should prove the convergence of a certain sequence of random measures $\nu_n$ to the Levy measure of the infinitely divisible law, and then show that this limiting measure is the Lévy measure of a Stable law.

Another interesting research direction consists in studying the training dynamics of Stable NNs. For Gaussian NNs, Jacot et al. (2018) and Arora et al. (2019) established the equivalence between a specific training setting of deep Gaussian NNs and kernel regression. In particular, they considered a deep Gaussian NN where the hidden layers are trained jointly under quadratic loss and gradient flow, i.e. gradient descent with infinitesimal learning rate, and it was shown that, as the width of the NN goes to infinity simultaneously, the point predictions are arbitrarily close to those given by a kernel regression with respect to the so-called neural tangent kernel. Such an analysis is typically referred to as the neural tangent kernel analysis of the NN (Arora et al., 2019). The large-width training dynamics of shallow Stable NNs with ReLU activation function, and input $\mathbf{X}$, has been considerd in Favaro et al. (2022b). In particular, they proved linear convergence of the

squared error loss for a suitable choice of the learning rate. The equivalence between gradient flow and kernel regression is connected to the so-called "lazy training" phenomenon, which is one of the hottest topics in the field of machine learning since it is a phenomenon which can affect any model, not only NNs. More precisely, Chizat et al. (2019) showed that lazy training is caused by an implicit choice of the scaling and that every parametric model can be trained in the lazy regime provided that its output is initialized close to zero. Furthermore, coming back to NNs, they considered a two layers NN with Gaussian weights and proved a sufficient condition for achieving lazy training, provided that $\mathbb{E}[w_i^{(0)}\tau(\langle w_i \cdot x\rangle)] = 0$. Clearly, the theorem applies also in the case of symmetric Stable weights and biases when $\alpha > 1$, but not when $\alpha \leq 1$ as the expectation of such RVs is undefined. It would be then interesting to study what happens in that case in order to find a new theoretical result which leads to a suitable scaling for which we have the lazy training regime.

## Acknowledgements

The authors are very grateful to the Action Editor (Jeffrey Pennington) and three Referees for their comments and suggestions that improved remarkably the paper. Stefano Favaro received funding from the European Research Council (ERC) under the European Union's Horizon 2020 research and innovation programme under grant agreement No 817257. Stefano Favaro is also affiliated to IMATI-CNR "Enrico Magenes" (Milan, Italy).

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

## A  Complementary statements and proofs

**Definition A.1** (Multivariate Stable distribution)**.** *Let $\mathbb{S}$ be the unit sphere in $\mathbb{R}^k : \mathbb{S} = \left\{ u \in \mathbb{R}^k : |u| = 1 \right\}$. A random vector, $X$, has a multivariate Stable distribution, denoted as $X \sim St_k(\alpha, \Gamma, \delta)$, if the joint characteristic function of $X$ is*

$$\mathbb{E}[\exp\left(iu^T X\right)] = \exp\left\{ -\int_{s \in \mathbb{S}} \left\{ \left|u^T s\right|^\alpha + i\nu\left(u^T s, \alpha\right) \right\} \Gamma(ds) + iu^T \delta \right\}$$

*where $0 < a < 2$, and for $y \in \mathbb{R}$*

$$\nu(y, \alpha) = \begin{cases} -\operatorname{sign}(y)\tan(\pi\alpha/2)|y|^\alpha & \alpha \neq 1 \\ (2/\pi)y\ln|y| & \alpha = 1 \end{cases}$$

*The case with $\delta = 0$ is denoted by $St_k(\alpha, \Gamma)$.*

The next two theorems characterize the infinitely wide limits of a deep NN with Gaussian and Stable parameters respectively, under the setting of joint growth and taking a matrix as input.

**Theorem A.1** ((de G. Matthews et al., 2018))**.** *For any $d \geq 1$ and $k \geq 1$ let $\mathbf{X}$ denote a $d \times k$ (input signal) matrix, with $\mathbf{x}_j$ being the $j$-th (input signal) row, and for any $D \geq 1$ and $n \geq 1$ let: i) $\left(\mathbf{W}^{(1)}, \ldots, \mathbf{W}^{(D)}\right)$ be i.i.d. random (weight) matrices, such that $\mathbf{W}^{(1)} = \left(w_{i,j}^{(1)}\right)_{1 \leq i \leq n, 1 \leq j \leq d}$ and $\mathbf{W}^{(l)} = \left(w_{i,j}^{(l)}\right)_{1 \leq i \leq n, 1 \leq j \leq n}$ for $2 \leq l \leq D$, where the $w_{i,j}^{(l)}$'s are i.i.d. as $N\left(0, \sigma_w^2\right)$ for $l = 1, \ldots, D$; ii) $\left(\mathbf{b}^{(1)}, \ldots, \mathbf{b}^{(D)}\right)$ be i.i.d. random (bias) vectors, such that $\mathbf{b}^{(l)} = \left(b_1^{(l)}, \ldots, b_n^{(l)}\right)$ where the $b_i^{(l)}$'s are i.i.d. as $N\left(0, \sigma_b^2\right)$ for $l = 1, \ldots, D$. Now, let $\phi : \mathbb{R} \to \mathbb{R}$ be a continuous activation function (nonlinearty) such that*

$$|\phi(s)| \leq a + b|s|$$

*for every $s \in \mathbb{R}$ and for any $a, b \geq 0$, and consider a (fully connected) feed-forward $NN \left( f_i^{(l)}(\mathbf{X}, n) \right)_{1 \leq i \leq n, 1 \leq l \leq D}$ of depth $D$ and width $n$ defined as follows*

$$f_i^{(1)}(\mathbf{X}) = \sum_{j=1}^{d} w_{i,j}^{(1)} \mathbf{x}_j + b_i^{(1)} \mathbf{1}^T$$

*and*

$$f_i^{(l)}(\mathbf{X}, n) = \frac{1}{\sqrt{n}} \sum_{j=1}^{n} w_{i,j}^{(l)} \left( \phi \circ f_j^{(l-1)}(\mathbf{X}, n) \right) + b_i^{(l)} \mathbf{1}^T,$$

*with $f_i^{(1)}(\mathbf{X}, n) = f_i^{(1)}(\mathbf{X})$, where 1 is the k-dimensional unit (column) vector, and o denotes the element-wise application. For any $l = 1, \ldots, D$, if $\left( f_i^{(l)}(\mathbf{X}, n) \right)_{i \geq 1}$ is the sequence obtained by extending $\left( \mathbf{W}^{(1)}, \ldots, \mathbf{W}^{(D)} \right)$ and $\left( \mathbf{b}^{(1)}, \ldots, \mathbf{b}^{(\bar{D})} \right)$ to infinite i.i.d. arrays, then as $n \to +\infty$ jointly over the first l NN 's layers*

$$\left( f_i^{(l)}(\mathbf{X}, n) \right)_{i \geq 1} \xrightarrow{w} \left( f_i^{(l)}(\mathbf{X}) \right)_{i \geq 1},$$

*where $\left( f_i^{(l)}(\mathbf{X}) \right)_{i \geq 1}$ is distributed as the product measure $\otimes_{i \geq 1} N_k \left( \mathbf{0}, \Sigma^{(l)} \right)$, and the covariance matrix $\Sigma^{(l)}$ has the $(u, v)$-th entry defined recursively as follows:*

$$\Sigma_{u,v}^{(1)} = \sigma_b^2 + \sigma_w^2 \langle \mathbf{x}_u, \mathbf{x}_v \rangle$$

*and*

$$\Sigma_{u,v}^{(l)} = \sigma_b^2 + \sigma_w^2 \mathbb{E}[\phi(f)\phi(g)],$$

*where $(f, g) \sim N_2 \left( \begin{pmatrix} 0 \\ 0 \end{pmatrix}, \begin{pmatrix} \Sigma_{u,u}^{(l-1)} & \Sigma_{u,v}^{(l-1)} \\ \Sigma_{v,u}^{(l-1)} & \Sigma_{v,v}^{(l-1)} \end{pmatrix} \right).$*

**Theorem A.2** (Theorem 2 of Favaro et al. (2022a)). *For any $d \geq 1$ and $k \geq 1$ let $\mathbf{X}$ denote a $d \times k$ (input signal) matrix, with $\mathbf{x}_j$ being the j-th (input signal) row, and for any $D \geq 1$ and $n \geq 1$ let: i) $\left( \mathbf{W}^{(1)}, \ldots, \mathbf{W}^{(D)} \right)$ be i.i.d. random (weight) matrices, such that $\mathbf{W}^{(1)} = \left( w_{i,j}^{(1)} \right)_{1 \leq i \leq n, 1 \leq j \leq d}$ and $\mathbf{W}^{(l)} = \left( w_{i,j}^{(l)} \right)_{1 \leq i \leq n, 1 \leq j \leq n}$ for $2 \leq l \leq D$, where the $w_{i,j}^{(l)}$ 's are i.i.d. as $S_\alpha(\sigma_w)$ for $l = 1, \ldots, D$; ii) $\left( \mathbf{b}^{(1)}, \ldots, \mathbf{b}^{(D)} \right)$ be i.i.d. random (bias) vectors, such that $\mathbf{b}^{(l)} = \left( b_1^{(l)}, \ldots, b_n^{(l)} \right)$ where the $b_i^{(l)}$ 's are i.i.d. as*

$$S_\alpha(\sigma_b)$$

*for $l = 1, \ldots, D$. Now, let $\phi : \mathbb{R} \to \mathbb{R}$ be a continuous activation function (nonlinearty) such that*

$$|\phi(s)| \leq \left( a + b|s|^\beta \right)^\gamma$$

*for every $s \in \mathbb{R}$ and for any $a, b > 0, \gamma < \alpha^{-1}$ and $\beta < \gamma^{-1}$, and consider a (fully connected) feed-forward $NN \left( f_i^{(l)}(\mathbf{X}, n) \right)_{1 \leq i \leq n, 1 \leq l \leq D}$ of depth $D$ and width $n$ defined as follows*

$$f_i^{(1)}(\mathbf{X}) = \sum_{j=1}^{d} w_{i,j}^{(1)} \mathbf{x}_j + b_i^{(1)} \mathbf{1}^T$$

*and*

$$f_i^{(l)}(\mathbf{X}, n) = \frac{1}{n^{1/\alpha}} \sum_{j=1}^{n} w_{i,j}^{(l)} \left( \phi \circ f_j^{(l-1)}(\mathbf{X}, n) \right) + b_i^{(l)} \mathbf{1}^T$$

with $f_i^{(1)}(\mathbf{X}, n) = f_i^{(1)}(\mathbf{X})$, where $\mathbf{1}$ is the $k$-dimensional unit (column) vector, and $o$ denotes the element-wise application. For any $l = 1, \ldots, D$, if $\left( f_i^{(l)}(\mathbf{X}, n) \right)_{i \geq 1}$ is the sequence obtained by extending $\left( \mathbf{W}^{(1)}, \ldots, \mathbf{W}^{(D)} \right)$ and $\left( \mathbf{b}^{(1)}, \ldots, \mathbf{b}^{(\bar{D})} \right)$ to infinite i.i.d. arrays, then as $n \to +\infty$ jointly over the first $lNN'$ s layers

$$\left( f_i^{(l)}(\mathbf{X}, n) \right)_{i \geq 1} \xrightarrow{w} \left( f_i^{(l)}(\mathbf{X}) \right)_{i \geq 1}$$

where $\left( f_i^{(l)}(\mathbf{X}) \right)_{i \geq 1}$ is distributed as the product measure $\otimes_{i \geq 1} \mathrm{St}_k \left( \alpha, \Gamma^{(l)} \right)$, with $\alpha \in (0, 2)$, and the spectral measure $\Gamma^{(l)}$ being defined recursively as follows:

$$\Gamma^{(1)} = \left\| \sigma_b \mathbf{1}^T \right\|^\alpha \zeta_{\frac{\mathbf{1}}{\|\mathbf{1}^T\|}} + \sigma_w^\alpha \sum_{j=1}^d \|\mathbf{x}_j\|^\alpha \zeta_{\frac{\mathbf{x}_j}{\|\mathbf{x}_j\|}}$$

and

$$\Gamma^{(l)} = \left\| \sigma_b \mathbf{1}^T \right\|^\alpha \zeta_{\frac{\mathbf{1}^T}{\|\mathbf{1}^T\|}} + \int \left\| \sigma_w(\phi \circ f) \right\|^\alpha \zeta_{\frac{\phi \circ f}{\|\phi \circ f\|}} q^{(l-1)}(df)$$

where

$$\zeta_{\frac{h}{\|h\|}} = \frac{1}{2} \begin{cases} \delta_{\frac{h}{\|h\|}} + \delta_{-\frac{h}{\|h\|}} & \text{if } \|h\| > 0 \\ 0 & \text{otherwise,} \end{cases}$$

with $\delta$ being the Dirac measure, and $q^{(l-1)}$ is the distribution of $f_i^{(l-1)}(\mathbf{X})$. The limiting $SP \left( f_i^{(l)}(\mathbf{X}) \right)_{i \geq 1}$ is referred to as the Stable SP with parameter $(\alpha, \Gamma)$.

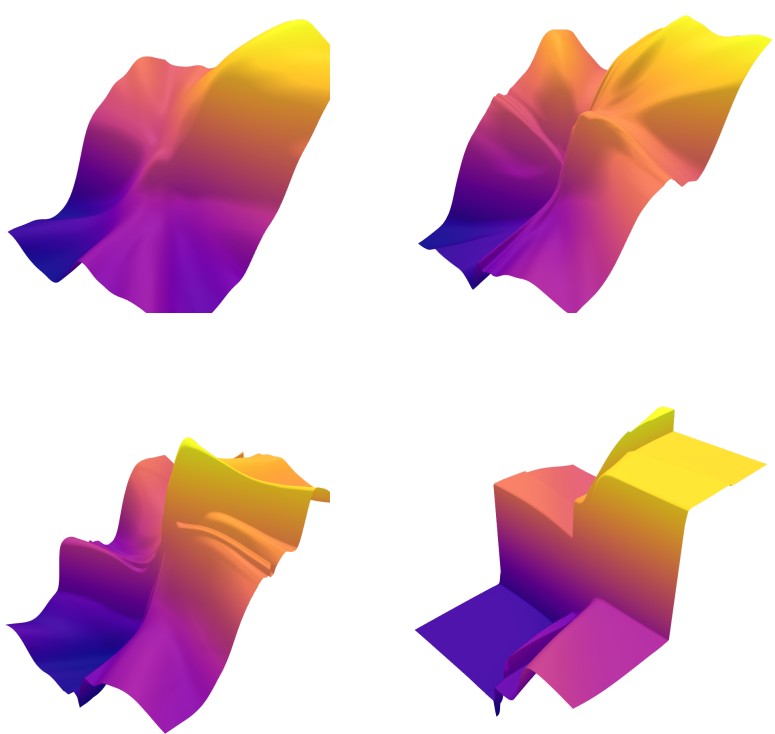

Figure 1: [Figure 1 of Favaro et al. (2022b)]. Samples of a shallow Stable NN mapping $[0,1]^2$ to $\mathbb{R}$, with a tanh activation function and witdth $n = 1024$, for different values of the stability parameter $\alpha$: i) $\alpha = 2.0$ (Gaussian distribution) top-left; ii) $\alpha = 1.5$ top-right; iii) $\alpha = 1.0$ (Cauchy distribution) bottom-left; iv) $\alpha = 0.5$ (Lévy distribution) bottom-right.

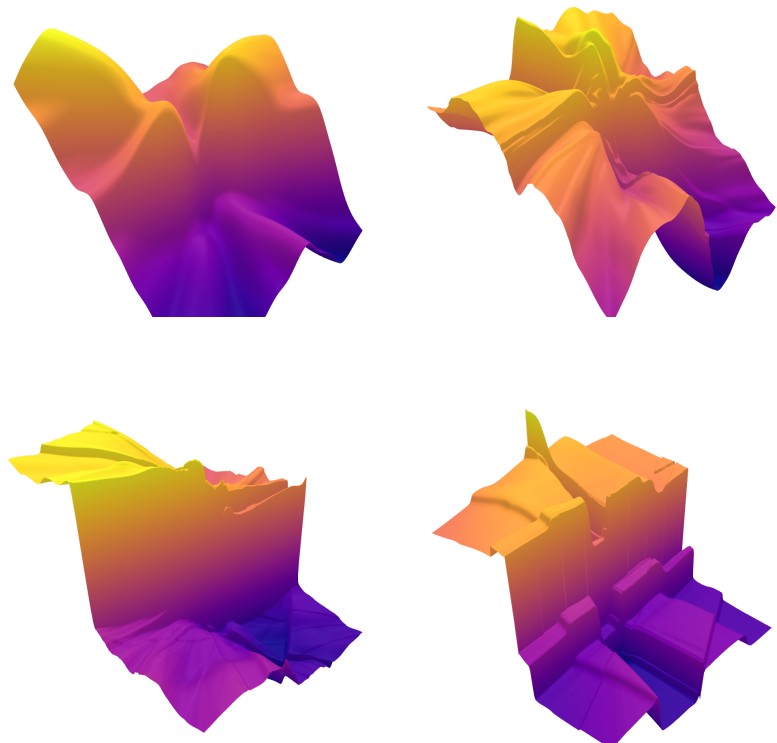

Figure 2: [Figure 1 of Favaro et al. (2022a)]. Samples of a deep Stable NN mapping $[0,1]^2$ to $\mathbb{R}$, with a tanh activation function and 2 hidden layers of width $n = 1024$, for different values of the stability parameter $\alpha$: i) $\alpha = 2.0$ (Gaussian distribution) top-left; ii) $\alpha = 1.5$ top-right; iii) $\alpha = 1.0$ (Cauchy distribution) bottom-left; iv) $\alpha = 0.5$ (Lévy distribution) bottom-right.

