# OpenReview forum: "Infinitely wide limits for deep Stable neural networks: sub-linear, linear and super-linear activation functions"
_TMLR — Accepted by TMLR_

### Review · Reviewer_zH9t · 2022-08-22

**Summary Of Contributions:**

It is well-established that, under mild assumptions, infinite width neural networks with well-behaved i.i.d. parameters converge in law to Gaussian processes as the width goes to infinity. This work studies a relaxation where the network's parameters belong to an alpha-stable distribution and the limiting stochastic process is a so-called Stable stochastic process.
- The authors study certain classes of sub-linear, linear and super-linear activations (E1, E2, E3, section 1.1).
- Unlike the Gaussian case where scaling is $1/\sqrt{n}$ for a width $n$, here the scaling may also involve an activation-dependent logarithmic factor.
- Theorem 2.6 gives the result for shallow networks.
- Theorem 2.7 gives the result for deep networks, E1, E2 and E3, sub-linear (roughly),
- Theorem 2.8 gives E2, E3 and super-linear

**Requested Changes:**

Minor points:
- the notation for the distributions in the first paragraph. $w \sim w \sim \mathcal{N}$. I think what we mean here is actually more like $w \stackrel{d}{=} w \sim \mathcal{N}$. The first notion of equality is between a random variable and a random variable, and the second is between a random variable and a distribution. Same for page 2.
- It is a little bit confusing to use $\sim$ to denote equal limits and distributed according to, e.g. in Theorem 2.1.
- It would be nice to have a definition of a slow varying function mentioned in Theorem 2.1, perhaps around the beginning of section 2?
- I am not familiar with the terminology "truncated CDF". Is this maybe supposed to be "complementary CDF"? Is truncated CDF also terminology in use?
- Top of page 6. "stated in the next theorem."
- "and it is one of the hottest topic in the field of machine learning since ". Change "topic" to "topics", or maybe even rejig this sentence so it is a bit less casual.
- Theorem 2.7 as a period . and comma , for $\gamma = 1$.

**Strengths And Weaknesses:**

Strengths:
- The flow of the presentation of theoretical results seems logical and sensible.
- This is a direction of infinite-width models that has not been explored a lot (in contrast with showing that another architecture converges to a GP, or finding the covariance function of another activation function, or finding good approximation algorithms, or etc.)
- The proofs appear to be a straight-forward application of the GCLT. The construction is where the innovation happens.

Weaknesses:
- No quantification of the rate of convergence
- There are no experiments in this paper. While this is not a requirement per se, it would be nice to see. For example, you could evaluate the MMD between a finite width process and an infinite width process. Even more compelling would be if you could do inference using one of these infinite width models.

Questions:
- In the GCLT, I am trying to understand the centering in the case $p =1$. When $p \neq 1$, the centering term does not depend on $n$. But when $p=1$, it seems to get larger in magnitude with $n$. Is this correct? Why does this happen for $p=1$ but not the others?
- (An interesting question IMO, though not necessarily one that needs to be answered in this paper. Maybe discussed though.) Is the additional logarithmic scaling encountered in certain settings "meaningful"? Maybe around the Chizat and Bach reference in the conclusion. Can finite-width models be trained lazily in this regime?

---

> ### Author Response · Authors · 2022-09-03
> **Reply to the Referee's report (zH9t)**
>
> We wish to thank the Referee zH9t for spending her/his time in reading our work, and for all the interesting comments. Below we discuss the points of weakness of our work, and reply to the questions raised in the report.
>
>
> WEAKNESSES
>
> Rates of convergence. The study of rates of convergence is certainly an interesting problem. To the best of our knowledge, the work of Basteri and Trevisan (2022) - arXiv:2203.07379 - is the first to provide a comprehensive analysis of the rates of convergence, in Wasserstein distance, for deep Gaussian NNs, under some suitable assumptions of smoothness for the activation function. The work of Favaro et al. (2022) - arXiv:2108.02316 - first investigated rates of convergence, in the sup-norm distance, for deep Stable NNs with a sub-linear activation function. As highlighted in these very recent papers, the study of rates of convergences, both in the Gaussian setting and the Stable setting, is not an easy task within the context of deep NNs, especially when it is assumed that the NN has $k>1$ input of dimension $d\geq1$. Motivated by the results of our work, it is our plan for future research to extend Theorem 11 and Theorem 12 of Favaro et al. (2022) to the context of deep Stable NNs with a ReLU activation function. We believe this is possible, though we expect to not be an easy task.
>
> Numerical experiments. Regarding the evaluation of the MMD between the deep Stable NN and its infinitely wide limit, an analogous numerical experiment is presented in the supplementary material of the work of Favaro et al. (2020) - arXiv:2003.00394 - for deep Stable NNs with a sub-linear activation function. It is possible to develop the same experiments with respect to the more general results presented in our work, i.e. under a ReLU activation function and super-linear activation functions. Regarding the use of the infinitely wide limit to make inference in the Stable setting, this is an interesting problem. Such a problem is discussed in the work of Favaro et al. (2022) - arXiv:2108.02316 - for deep Stable NNs with a sub-linear activation function. In particular, for a NN with $k>1$ inputs of dimension $d\geq1$, Favaro et al. (2022) discussed how making inference with the infinitely wide Stable process is more challenging than making inference with the infinitely wide Gaussian process. The main issue is due to the computational cost in the evaluation of the distribution of the infinitely wide Stable process, which requires a discretization of the corresponding spectral measure.
>
>
> QUESTIONS
>
> Question 1. In the case $p<1$ no centering turns out to be necessary, due to the "large" normalizing constants $n^{-1/p}$, which smooth out the differences between the right and the left tail. In the case $p>1$, the centering is the common one, that is the expectation. The case $p=1$ is indeed a particular case: the expectation does not exist, so it cannot be used for centering; on the other hand, centering is necessary for convergence because the normalizing constant $n^{-1}$ does not grow sufficiently fast to smooth the differences between the right and the left tail. The term including $\log n$ comes from the asymptotic behaviour of truncated moments. We have clarified this point in the revised paper.
>
> Question 2. As we have discussed in Section 3, the work of Favaro et al. (2022) - arXiv:2206.08065 - provides the neural tangent kernel analysis of a shallow Stable NN with a ReLU activation function. In particular, Favaro et al. (2022) characterized the large-width training dynamics of the Stable NN with respect to a kernel regression with a Stable random kernel. This analysis shows a “lazy training” phenomenon for shallow Stable NN rescaled by $(n\log n)^{-1/\alpha}$. In general, for NN’s weights distributed according to a distribution with finite moments, Chizat et al. (2019) provided a sufficient condition for the “lazy training” phenomenon to hold, and showed that Gaussian NNs rescaled by $n^{-1}$ do not satisfy such a conditions. In particular, the scaling $n^{-1}$ allows the NN’s dynamics to converge to a non degenerate dynamic described by a partial differential equation and referred to as the mean-field limit (Chizat and Bach, 2018; Rotskoff and Vanden-Eijnden, 2018; Mei et al., 2018; Sirignano and Spiliopoulos, 2020). Then, one may consider the problem of finding a sufficient condition for the “lazy training” phenomenon in the Stable setting, i.e. the infinite variance setting, and then look for possible scaling that violate such a condition. Work on this research direction is ongoing.
>
> REQUESTED CHANGES
>
> Thank you for the comments. We have addressed all the minor points in the revised version of the paper.

---

### Review · Reviewer_WgWA · 2022-08-23

**Summary Of Contributions:**

The paper studies infinite-width limits of (deep) neural networks with weights sampled from $\alpha$-Stable distributions (which may have different tail behaviors than the usual Gaussian initialization, for which $\alpha = 2$), and actions with different asymptotic growth conditions.
The authors provide convergence results to stable distributions for shallow and deep networks with different stable weight distributions and activations, along with appropriate scaling factors, when looking at the network output on a single input example. In particular, for asymptotically linear activations such as the ReLU, it is shown that the scaling factor needs an additional logarithmic term, and for activations with super-linear growth, the output distribution may have a scale that shrinks exponentially with depth, when all layers have an equal scale.

**Requested Changes:**

Despite the limited significance, the results in the present paper may be useful for the community. Here are some suggestions for how the paper could be improved:
* more discussion of the motivation for studying Stable networks, as explained in the previous section.
* a more detailed comparison with related works, in particular Favaro et al., 2022, which considers the specific case of the ReLU activation.
* in terms of presentation, perhaps the examples in section 2.3 could be incorporated in the corresponding prior sections?

Some additional minor comments/typos:
* the term "linear activation" e.g. in the abstract sounds like you're only considering linear activation functions, although this is only referring to the asymptotic behavior. Perhaps you can change this to "asymptotically linear" for more generality?
* typo at end of abstract: "infinitely with"?
* section 1.2: "shallow Stable with" -> "shallow Stable NN with"?
* beginning of proof of Lemma 2.4: $z$ should be $x$?
* before Thm 2.6: "CLT 2" refers to Theorem 2.1?

**Strengths And Weaknesses:**

Strengths: the results are quite general, supporting a broad class of activations, as well as large choices of stable distributions for the weights.

Weaknesses:
- The improvements over previous works of Favaro et al. are somewhat minor, mainly regarding extending the generality of activations. For instance, the presence of the log-factor in the scaling seemed to already be present in [this previous paper](https://arxiv.org/abs/2206.08065) for the special case of the ReLU.
- The results only study convergence of the network output for a single fixed input, as opposed to previous works which study additional properties of the stochastic process, namely the dependency structure across different inputs, e.g. through a covariance function (in the GP case), or spectral measure on a finite set of datapoints (in the Stable case).
- The motivation for studying Stable weights instead of Gaussian is somewhat limited. It would be helpful to provide more references that highlight their relevance. (looking at some of the papers by Favaro et al., it seems that an additional, interesting motivation is that the weights *after training* often exhibit heavy tails, even though the initialization is Gaussian)

---

> ### Author Response · Authors · 2022-09-03
> **Reply to the Referee's report (WgWA)**
>
> We wish to thank the Referee WgWA for spending her/his time in reading our work, and for all the interesting comments. Below we discuss the points of weakness of our work, and reply to the questions raised in the report.
>
> WEAKNESSES
>
> Improvements over previous works, and the use of a single fixed index. The aim of our work is to bring out the importance of the generalized central limit theorem (GCLT), which is not as popular as the classical central limit theorem, in the study of the large width behaviour of deep Stable NNs. As the classical central limit theorem plays a critical role in the study of the large-width behaviour of deep Gaussian NNs (Neal, 1994), our work shows how the GCLT plays the same critical role in the study of the large-width behaviour of deep Stable NNs. In particular, the use of GCLT allows to reduce the problem of characterizing the large-width behaviour of deep Stable NNs to the problem of characterizing the tail behaviour of the product of suitable transformations, depending on the activation function, of Stable random variables. Our study leads to a unified treatment of deep Stable NNs with an input of dimension $d\geq1$ and, as we have discussed in Section 3, it poses the problem of introducing a multidimensional version of the GCLT in order to deal with deep Stable NNs with $k>1$ inputs of dimension $d\geq1$. To the best of our knowledge, multidimensional versions of the GCLT, i.e. dealing with $k>1$ dimensional Stable distributions, are not available in the probabilistic/statistical literature, and they may be of independent interest.
>
> The work of Favaro et al. (2022) - arXiv:2108.02316 - characterizes the large-width behaviour for deep Stable NNs with a sub-linear activation function, assuming $k>1$ inputs of dimension $d\geq1$, whereas the work of Favaro et al. (2022) - arXiv:2206.08065 - characterizes the large-width behaviour for a shallow Stable NNs with a ReLU activation function, still assuming $k>1$ inputs of dimension $d\geq1$. In these papers, the assumption of $k>1$ inputs led to develop ad hoc proofs for characterizing the large-width behaviours of the NNs, as a $k$-dimensional version of the GCLT is not available from the literature. These proofs are based on a careful study of the characteristic function of the NNs, and then they  characterize the large-width behaviours of the NNs in terms of the infinitely wide limits for their corresponding characteristic functions.
>
>
> Motivation. Because of the constraint on the number of pages, we reported what is arguably the original motivation for replacing the Gaussian distribution of the NN’s parameters with a general Stable distribution, and then referred to Favaro et al. (2022) - arXiv:2108.02316 - for more details. As briefly described in our work, the original motivation for the use of Stable distributions comes from empirical analyses in the seminal work of Neal (1996), which show that while the contribution of Gaussian weights vanishes in the infinitely wide limit, Stable weights retain a non-negligible contribution. See also Der and Lee (2006) and Lee et al. (2022).  Most recently, empirical analyses in Fortuin et al. (2019) showed that wide Stable NNs trained with gradient descent lead to a higher classification accuracy than Gaussian NNs (Fortuin, 2021 and Lee et al., 2022). Such different behaviours between the Gaussian setting and the Stable settings arise from different large-width sample path properties of the NNs, as shown in Figure 1 of Favaro et al. (2022) , which make Stable NNs more flexible than Gaussian NNs.
>
>
> REQUESTED CHANGES
>
> Thank you for the comments. In the revised paper, we have rewritten some parts of Section 1 in order to: i) better clarify the aim of this work, especially with respect to the previous works on the large-width behaviour of deep Stable NNs; ii) improve and expand the motivation for the use of Stable distributions in order to initialize deep NNs. In relation to that, we also have rewritten some parts of Section 3. In doing this changes, we did not change the length of the paper, staying in the 12 pages. We have addressed all the minor points in the revised version of the paper.

---

### Review · Reviewer_X1ij · 2022-09-10

**Summary Of Contributions:**

The paper examines large width function space behaviour of fully connected neural networks (FCNs) with weights drawn i.i.d. from a **stable**—instead of Gaussian—distribution. Some results of this kind have already been obtained in the cited papers by Favaro et al. (2020, 2021). One limitation of the Favaro et al. papers is that they assume that the  activation function $\tau$ is bounded by a  **strictly** sub-linear function. This among else excludes linear and ReLU-like non-linearities. (The sublinearity in Favaro et al. is a simple way to ensure that $\tau(X)$ has well-behaved tail behaviour for $X$ drawn from a symmetric stable distribution $S_\alpha(\sigma)$.) The paper presented here extends the class of activation functions to those bounded by certain linear and super-linear functions. This extends the class of non-linearities for which we understand large width behaviour to, e.g., ReLU and polynomial activations, but—unlike in the Favaro et al. (2020, 2021)—restricts focus onto **1-hidden layer** FCNs. (Only _sequential limits_ considered for deeper FCNs; more on this in "Strengths and Weakness").

To me, the main contribution of the paper is in characterising the tail behaviour of $X \cdot \tau(Y)$ where $X \sim S_{\alpha} (\sigma)$ and $Y \sim S_{\alpha} (\sigma)$, for the various activation functions $\tau$, and subsequent application of these results to characterising the large width behaviour of 1-hidden layer FCNs at initialisation.

**Requested Changes:**

### Joint limit
I am concerned that the extension to linearly and super-linearly bounded activations, _while reverting to 1-hidden layer NNs_, is too thin to warrant an independent publication (esp. given the mentioned work of Favaro et al., which does prove the joint limit for strictly sub-linear activations).

A change that would convince me to recommend accept is proving the joint limit for the activations studied here.

I am potentially willing to reconsider the above requirement if the authors can provide a convincing argument for why proving the joint limit is beyond the current capabilities of the research community. Few (seemingly?) relevant papers:

* https://www.jstor.org/stable/2243946
* https://link.springer.com/chapter/10.1007/978-3-0348-8829-5_8
* https://arxiv.org/abs/1204.4357
* https://epubs.siam.org/doi/abs/10.1137/S0040585X97975459


### Relevance of stable limits

While I appreciate there is some theoretical interest in understanding the stable limits, there is little discussion of their practical importance (beyond a short mention of Radford Neal's original work).

It is not necessary to cover this in detail, but the work would be strengthened if the authors could provide a convincing argument (or even better, empirical evidence) for why stable limits could provide modelling benefits (as opposed to, e.g., the NNGP/NTK limits).

**Strengths And Weaknesses:**

### Strengths
* Extension of stable behaviour to a class of non-linearities which includes ReLU-like, polynomial, and linear FCNs.
* Interesting results on tail behaviour of products $X \cdot \tau (Y)$ where $X$ and $Y$ are independent symmetric stable random variables.

### Weaknesses
* Use of "sequential limit" to make statements about behaviour of FCNs with more than one hidden layer.
* Little discussion the "joint limit", and the difficulties of deriving it for stable init/why it is not included in the current paper.

Here:
- "Sequential limit" characterises large width behaviour of **1-hidden layer** NNs, optionally with random inputs drawn from a stable distribution.
- "Joint limit" characterises large width behaviour of arbitrary fixed depth NN, where all layers are finite and grow wide together.

This distinction is not sufficiently emphasised, and the theorems claiming to characterise behaviour of **deep** stable NNs are thus misleading.

---

> ### Author Response · Authors · 2022-09-13
> **Reply to the Referee's report (X1ij)**
>
> We wish to thank the Referee X1ij for spending her/his time in reading our work, and for all the interesting comments. Below we discuss the points of weakness of our work, and reply to the questions raised in the report.
>
> WEAKNESS 1 (Joint limit)
>
> To the best of our knowledge, most of the literature considers the large $n$ asymptotic behavior of sums of normalized and centered exchangeable random variables: $Y_i^{(n)}=a_n \sum_{j=1}^n(X_{ij}-b_n)$, with the $(X_{ij})$'s being exchangeable random variables. This is treated in: i) https://www.jstor.org/stable/2243946; ii) https://link.springer.com/chapter/10.1007/978-3-0348-8829-5_8; iii) https://arxiv.org/abs/1204.4357. See also a section of https://epubs.siam.org/doi/abs/10.1137/S0040585X97975459. The results contained in these papers do not apply to the problem of characterizing the infinitely wide limit of deep Stable NNs in which the width grows jointly over the NN's layers. In such a context, the central limit theorem that would be requested would consider sum of the following type: $Z_i^{(n)}=a_n\sum_{j=1}^n (X_{ij}^{(n)}-b_n)$, with the $X_{ij}^{(n)}$ being exchangeable random variables for every $n$, but with different laws for different values of $n$. This case is substantially more difficult to treat.
>
> As far as we know the only available result that might be useful to study the large $n$ behaviour of $Z_i^{(n)}$ is  Theorem 1 in the paper https://epubs.siam.org/doi/abs/10.1137/S0040585X97975459. Such a theorem gives sufficient conditions for the convergence to a mixture of infinitely divisible laws. Applying this theorem seems infeasible to us, since checking the conditions would request the knowledge of the directing random measure of $X_{ij}^{(n)}$ for every $n$. Such a random measure has not a manageable closed form in the setting of deep Stable NNs.  As an alternative strategy, Favaro et al. (2021) developed an ad-hoc inductive approach and then characterized the large-width behaviours of the deep Stable NN in terms of the infinitely wide limit of its characteristic function. However, critical to the strategy of Favaro et al. (2021) is the assumption of a sub-linear activation function. Extending the inductive approach to linear and super-linear activation function seems infeasible to us.
>
> We are not aware of any paper in the literature containing results may be "concretely" helpful in the study of the the infinitely wide "joint limit" of deep Stable NNs with linear and super-linear activation functions.
>
> We would like to conclude with a general remark on the "joint limit" and the "sequential limit". As proved in Favaro et al. (2021), the "joint limit" and the "sequential limit" lead to the same infinitely wide limit for a deep Stable NN with a sub-linear activation function. This suggests that the difference between  the "joint limit" and the "sequential limit" can not be appreciated in a "qualitative" result such as the characterization of the infinitely wide limit of the NN. We conjecture that an analogous phenomenon holds true for deep Stable NNs with linear and super-linear activation functions, namely the infinitely wide limits presented in our work hold true when the width grows jointly over the layers. Differently from the study of the infinitely wide limit of the NN, results in Favaro et al. (2021) suggest that a critical difference between the "joint limit" and the "sequential limit" arises in a "quantitative" result such as the characterization of the rate of convergence of the NN to its infinitely wide limit. In particular, with respect to the sup-norm, Favaro et al. (2021) proved that the "joint limit" leads to a rate that depends on the depth, whereas the "sequential limit" leads to a rate that is independent of the depth.
>
> WEAKNESS 2 (Motivation)
>
> As briefly described in our work, the original motivation for the use of Stable distributions comes from empirical analyses in the seminal work of Neal (1996), which show that while the contribution of Gaussian weights vanishes in the infinitely wide limit, Stable weights retain a non-negligible contribution (Der and Lee, 2006 and Lee et al., 2022)  Most recently, empirical analyses in Fortuin et al. (2019) showed that wide Stable NNs trained with gradient descent lead to a higher classification accuracy than Gaussian NNs (Fortuin, 2021 and Lee et al., 2022). Such different behaviours between the Gaussian and the Stable settings arise from different large-width sample path properties of the NNs, as shown in Figure 1 of Favaro et al. (2022), which make Stable NNs more flexible than Gaussian NNs.
>
>
> REQUESTED CHANGES
>
> In the revised version on the paper, which has been already uploaded to address the comments of two Referees, we improved and expanded the motivation for the use of Stable distributions to initialize NNs. In a following revision, we will discuss more the use of the "sequential limit" versus the use of the "joint limit", highlighting the main difficulties of the latter.

---

### Author Response · Authors · 2022-10-25
**Any more comments?**

Dear Reviewers

Thank you again for your careful read and detailed comments about our manuscript. We were just wondering whether you could kindly let us know whether you are satisfied by our responses, or whether you have any remaining questions which we might be able to address.

Sincerely, The authors

---

### Decision · Action_Editors · 2022-10-31

**Recommendation:** Accept with minor revision

**Comment:**

This paper examines the infinite width limit of neural networks with heavy-tailed stable distributions, as opposed to Gaussian distributions. It generalizes prior work to examine this limit for a broader class of activation functions, including functions with linear and super-linear growth. The proof requires taking the layers of the network to infinity in a sequential manner, rather than jointly, and at least one reviewer found this restriction to be a fairly significant limitation of the results. I would agree with the reviewer that there may be a real distinction between the two limits and more emphasis should be devoted to this question. Otherwise, the reviewers and I agree that the paper is of good quality with relevant results and merits publication at TMLR.

To more thoroughly address the question of sequential vs joint limits, there are two options. First, the authors could prove their perspective (put forward in the discussion) that the two limits yield results that are qualitatively equivalent. Of course, this proof may be well out of scope, as the authors suggest in their comments. Alternatively, the authors could revise some of the their claims so that the sequential nature of the result is prominent, with caveats added to make clear that other conclusions might be possible if the limit were taken jointly. This latter approach would involve minor edits and is the minimum revision necessary to more fully satisfy the TMLR standards for acceptance.

**Audience:**

Yes, this paper will be of interest to members of the TMLR audience.

**Claims And Evidence:**

No, the claims are not currently supported by clear and convincing evidence. The issue relates to the sequential vs joint limit, but is relatively minor and could be easily remedied with a minor revision.